# This Time is Different: An Observability Perspective on Time Series Foundation Models

**Ben Cohen**[*‡]  **Emaad Khwaja**[*‡]
**Youssef Doubli**[†‡]  **Salahidine Lemaachi**[†‡]  **Chris Lettieri**[†‡]  **Charles Masson**[†‡]
**Hugo Miccinilli**[†‡]  **Elise Ramé**[†‡]  **Qiqi Ren**[†‡]  **Afshin Rostamizadeh**[†‡]  **Jean Ogier du Terrail**[†‡]
**Anna-Monica Toon**[†‡]  **Kan Wang**[†‡]  **Stephan Xie**[†‡§]  **Zongzhe Xu**[†‡]  **Viktoriya Zhukova**[†‡]
**David Asker**[‡]  **Ameet Talwalkar**[‡§]  **Othmane Abou-Amal**[‡]
{ben.cohen, emaad, ameet, othmane}@datadoghq.com

## Abstract

We introduce TOTO, a time series forecasting foundation model with 151 million parameters. TOTO uses a modern decoder-only architecture coupled with architectural innovations designed to account for specific challenges found in multivariate observability time series data. TOTO's pre-training corpus is a mixture of observability data, open datasets, and synthetic data, and is 4-10× larger than those of leading time series foundation models. Additionally, we introduce BOOM, a large-scale benchmark consisting of 350 million observations across 2,807 real-world time series. For both TOTO and BOOM, we source observability data exclusively from Datadog's own telemetry and internal observability metrics. Extensive evaluations demonstrate that TOTO achieves state-of-the-art performance on both BOOM and on established general purpose time series forecasting benchmarks. TOTO's model weights ( `https://huggingface.co/Datadog/Toto-Open-Base-1.0`), inference code , and evaluation scripts (`https://www.https://github.com/DataDog/toto`), as well as BOOM's data and evaluation code (`https://www.https://huggingface.co/datasets/Datadog/BOOM`), are all available as open source under the Apache 2.0 License.

## 1   Introduction

Observability is the practice of collecting and analyzing data generated by distributed computer systems to detect, diagnose, and swiftly resolve performance and reliability issues [4]. A major component of observability is monitoring time series metrics; observability tools generate massive and diverse sets of metrics that reflect a system's operational health over time. These metrics encompass a wide variety of indicators—such as memory usage, CPU load, disk I/O, network throughput, hit counts, error rates, and latency—that each exhibit distinct behavioral patterns, and collectively represent an important but under-studied subset of general time series data.

Accurately modeling observability metrics is essential for critical tasks like anomaly detection [5] (e.g., identifying spikes in error rates) and predictive forecasting [6] (e.g., anticipating resource exhaustion or scaling needs). Observability data present challenges for traditional forecasting methods due to diversity, high-dimensionality, and complex distributional characteristics (Section 4.3). Moreover, real-world observability systems routinely generate millions to billions of distinct time

---

[*]Joint First Author, listed alphabetically

[†]Core Contributor, listed alphabetically

[‡]Datadog AI Research

[§]Carnegie Mellon University. AT and SX contributed to this work in their Datadog capacities.

39th Conference on Neural Information Processing Systems (NeurIPS 2025).

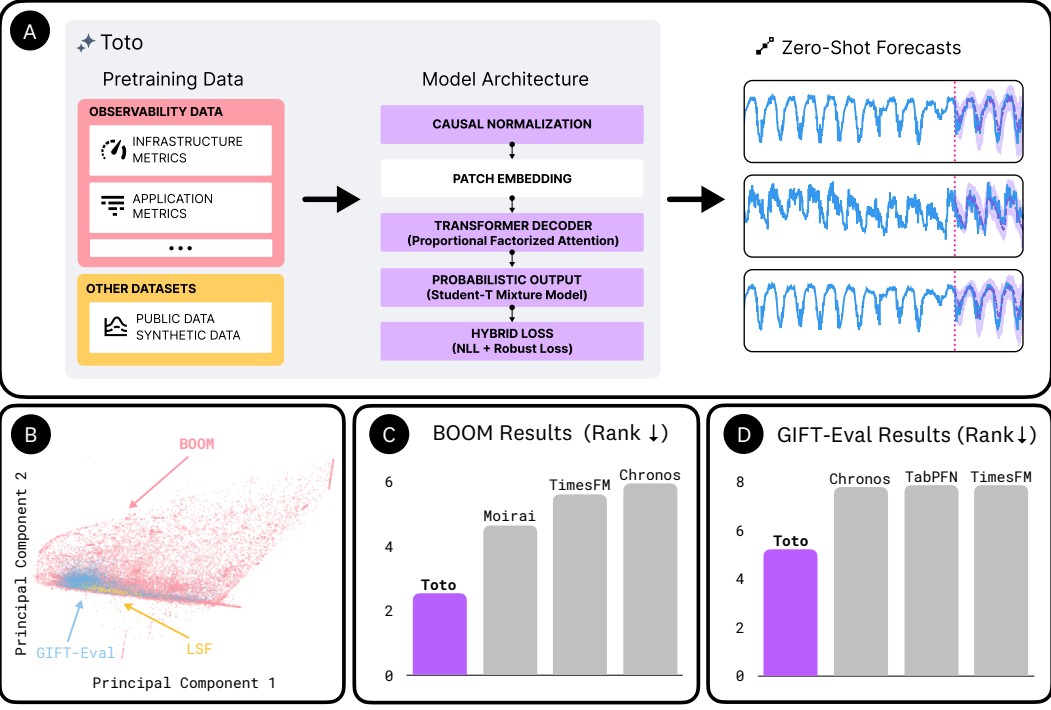

Figure 1: Ⓐ TOTO is a zero-shot time series forecasting model trained on a mixture of observability data, open datasets, and synthetic data. To predict, context time series points are passed through a patch embedding, processed via proportional factorized variate-time attention layers, and projected to a probabilistic output via a learned Student-T Mixture model. We sample from this distribution to produce a prediction forecast. Note that TOTO's novel architectural components are highlighted in purple. Ⓑ 2D PCA projections of statistical features (described in Section 4.3) of GIFT-Eval [1], LSF [2], and BOOM highlight a clear distinction in the underlying time series characteristics of BOOM relative to general-purpose time series benchmarks. Ⓒ, Ⓓ TOTO is the top performing model on BOOM, the GIFT-Eval public leaderboard [3], and on LSF (see Table 4).

series [7–9], rendering fine-tuning or supervised training of complex models per time series infeasible. These operational challenges suggest a compelling use case for zero-shot time series foundation models (FMs). However, we find that existing FMs [10–12] trained for general-purpose forecasting struggle to generalize to observability data; see Fig. 1C and Section 5.

In this work, we focus on the unique challenges of modeling observability data, while accounting for the constraints of production settings. Our two main contributions are as follows:

1. **TOTO** (**T**ime Series **O**ptimized **T**ransformer for **O**bservability). We develop a novel open-weights time series forecasting foundation model, with a focus on zero-shot capabilities. **TOTO uses a modern decoder-only architecture coupled with architectural innovations to account for the specific challenges found in observability time series data**: a novel per-variate patch-based causal scaling to address highly nonstationary sequences; proportional time-variate factorized attention to judiciously attend across a large number of covariates; and a Student-T mixture prediction head optimized via a robust composite loss to fit complex and highly skewed distributions. TOTO's pretraining corpus contains 4-10× more unique data points than those of other time series FMs [11–14], using a mix of domain-specific observability time series data, multi-domain public datasets, and synthetic data.

2. **BOOM** (**B**enchmark **o**f **O**bservability **M**etrics). We introduce an open-source benchmark specifically for observability time series. BOOM includes a large-scale, novel dataset with 350 million observations across 2,807 distinct multivariate time series, approximately twice the size of the general-purpose GIFT-Eval benchmark [3]. Unlike existing multi-domain benchmarks, **BOOM is comprised entirely of real-world observability data.** It captures diverse time series dynamics and challenging behaviors across several subdomains (Fig. 4), thus providing a uniquely realistic and comprehensive evaluation environment for forecasting models.

In our evaluations against leading foundation models and traditional time series forecasting base-lines, **TOTO achieves state-of-the-art performance on both general-purpose and observability-oriented time series forecasting benchmarks.** On BOOM, TOTO achieves a 12% improvement in terms of CRPS compared to the next best method (see Section 5). TOTO also achieves the top position by a wide margin on two standard general-purpose time series benchmarks—GIFT-Eval and Long Sequence Forecasting (LSF)—implying our observability-focused design also pays dividends in other time series domains [1, 15]. We additionally perform ablations to motivate TOTO's architecture design (Fig. 3B), and conduct data analyses of BOOM to illustrate the unique aspects of observability data that pose significant modeling challenges (Section 4).

For both TOTO and BOOM, we source observability data exclusively from our own telemetry and internal observability metrics. We provide TOTO's model weights, inference code, and evaluation scripts, as well as BOOM 's evaluation code and data, under a permissive (Apache 2.0) license.

## 2    Related Work

**Supervised models**. Current observability systems typically rely on classical modeling approaches such as Holt-Winters [5], tree-based methods [16], or (S)ARIMA [17] for forecasting and anomaly detection [18]. These approaches require individual training for each dataset, impeding scalability in large real-world systems. While neural models have been shown to surpass these classical models in some settings [19] they are generally larger, more complex, and still require training for each datset; they are thus operationally infeasible in practice.

**Time series foundation models**. By pre-training on large multi-domain datasets, several time series foundation models [11–14, 20–25] have achieved impressive zero-shot prediction capabilities on general purpose time series benchmarks, eliminating the need for domain-specific training or finetuning. This approach is especially promising for observability workloads, as a single model can be deployed and horizontally scaled to provide low-latency and relatively low-cost zero-shot inference. Our evaluations indicate that TOTO outperforms existing time series foundation models by a wide margin on both public forecasting benchmarks and on BOOM (Section 5).

**Time series benchmarks**. Traditional benchmarks include Monash [26], LSF [2], M3 [27], and M4 [28]. These benchmarks are either commonly used for pretraining FMs (Monash) [11, 13] or are limited in capacity in measuring the impact of modern methods (LSF, M3, M4) [29]. Aksu et al. [1], Qiu et al. [30], and Ansari et al. [11] all recently introduced multi-domain benchmarks that are better suited in scale and complexity to evaluate general-purpose time series foundation models. GIFT-Eval [1] in particular is orders of magnitude larger than LSF (see Table 1 for details); introduces standardized evaluation protocols to facilitate fair comparisons across models; and is unique in containing a large decontaminated pretraining dataset for rigorous measurement of zero-shot capabilities without test data leakage.

Our evaluations show that TOTO achieves state-the-art-performance on LSF and GIFT-Eval. Moreover, BOOM adopts the evaluation protocols introduced in GIFT-Eval, though it is unique in its scale (it has approximately twice as many time series points as GIFT-Eval) and its domain of focus (it is derived entirely from real-world observability data).

**Challenges of observability data**. Several works have explored observability time series data and the prospect of applying the foundation model paradigm to it. Joosen et al. [31] analyze serverless function logs from Huawei Cloud, highlighting challenges such as values spanning nine orders of magnitude, heavy-tailed distributions, and multi-scale seasonality. Toner et al. [32] find that existing time series foundation models perform poorly on a small dataset of cloud data. Woo et al. [33] pretrain models on CPU and memory metrics, while Palaskar et al. [34] propose an observability-specific architecture trained in a full-shot setting on four small-scale observability datasets. Both BOOM and TOTO's pretraining corpus are more diverse and in most cases orders of magnitude larger than the datasets considered in prior studies. TOTO is also the first observability-specialized foundation model that also performs competitively on broad benchmarks like GIFT-Eval and LSF.

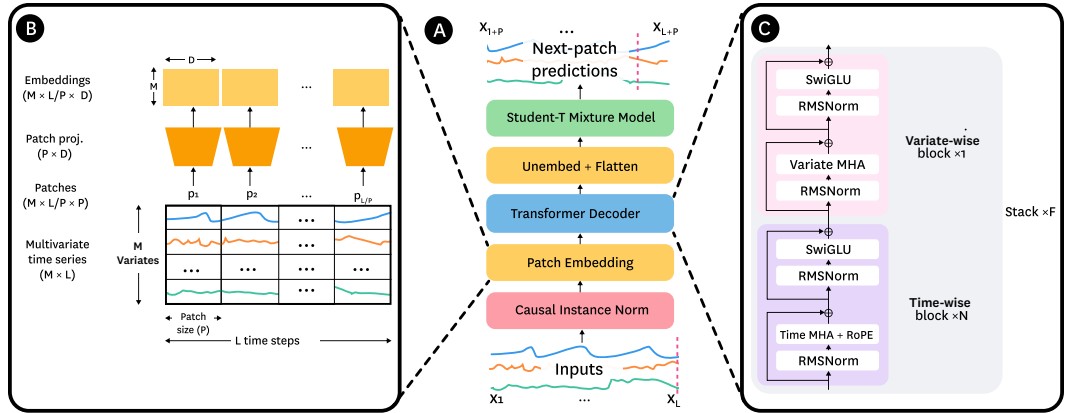

Figure 2: Overview of the TOTO architecture, highlighting our novel components in **bold**. (A) Multivariate input time series of $L$ steps are scaled using **causal patch-based instance normalization**, transformed into patch embeddings, and passed through a decoder-only transformer stack. The transformed features are unembedded and passed through a **Student-T mixture model** (optimized via a **composite robust loss**) which generates probabilistic next-patch predictions. (B) The patch embedding takes as input a time series of $M$ variates by $L$ time steps. It divides the time dimension into patches of size $P = 64$ and projects these linearly into an embedding space of latent dimension $D = 768$. This results in an output of size $M \times \frac{L}{P} \times D$ which is fed to the transformer decoder. (C) The transformer stack features **proportional factorized attention**. It contains $F = 1$ identical segment(s), with $N = 11$ time-wise transformer blocks followed by one variate-wise block.

# 3 TOTO

We next outline key architectural components of TOTO, and summarize its large-scale pre-training corpus. See Appendix A for extended details and full hyperparameters.

## 3.1 Model architecture

Transformer models for time series forecasting have variously used encoder-decoder [2, 11, 15], encoder-only [13, 19, 35], and decoder-only architectures [12, 21]. TOTO uses a decoder-only architecture (trained on a next-patch prediction task), as it has been shown to scale well with respect to training efficiency when provided with sufficient data [36, 37]. We use non-overlapping patch embeddings [19, 38, 39], with a patch of size $P = 64$, to project input time series of context length $L = 4096$ points to embeddings of size $64 \times D$ per variate, where $D = 768$ is the embedding dimension for our final model (Fig. 2B). We also utilize techniques demonstrated to yield performance and efficiency improvements in contemporary transformer literature, including pre-normalization [40], RMSNorm [41], SwiGLU feed-forward layers [42], and RoPE [43] with XPOS [44] for improved extrapolation.

We further develop four specialized components purpose-built for handling multivariate observability time series data. Fig. 3B presents an ablation study that demonstrates the impact of these components, and we next highlight the motivations and intuitions behind each of them.

**Patch-based causal instance normalization to handle highly nonstationary data**. To improve generalization across varying input scales, instance normalization is commonly applied prior to embedding time series data (for example, RevIN [45]). However, computing normalization statistics from the entire series would leak information from future time steps. This violates the causality of next patch prediction training and results in poor performance (see ablation in Appendix E). Das et al. [12] normalize the entire series according to the statistics of the first patch. Such an approach preserves causality, but can be ineffective for highly nonstationary data with statistics that vary significantly over time, as is the case with observability data (see Section 4.3).

To resolve these issues, we propose a novel per-patch normalization approach, where scaling factors for each patch are computed exclusively from the current patch and past data. For a timestep $t$, we

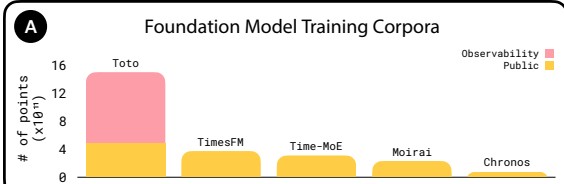
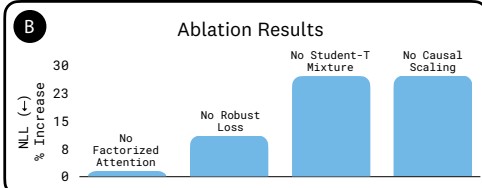

Figure 3: Ⓐ A comparison of the number unique time series points within the pretraining corpora of different time series foundation models. The scale of TOTO's training corpus is $4\times$ that of TimesFM 1.0, $5\times$ that of Time-MoE, $6.5\times$ that of Moirai, and over $10\times$ that of Chronos. Ⓑ Ablation results demonstrate the impact of four of TOTO's architectural components motivated by unique properties of observability time series data. Results report the change (relative to the full TOTO model) in negative log likelihood on held-out observability pretraining data when systematically disabling one component at a time. See Appendix E for details.

define the causal mean $\hat{\mu}_t$ and causal variance $\hat{s}_t$ as:

$$\hat{\mu}_t = \frac{\sum_{i=1}^{t} w_i x_i}{\sum_{i=1}^{t} w_i}, \quad \hat{s}_t = \sqrt{\frac{\sum_{i=1}^{t} w_i (x_i - \hat{\mu}_t)^2}{\sum_{i=1}^{t} w_i - 1} + 0.1},$$

where $x_i$ represents the input value and $w_i$ the corresponding weight at timestep $i$. We set the weight to 0 for padding positions and 1 for all other positions. We add a minimum value of 0.1 to the causal standard deviation, in order to limit the amount of scaling applied to any particular value and avoid numerical overflow. Timesteps within each patch share the normalization values determined by the final timestep of that patch. As computing causal statistics for every subsequence would have suboptimal $O(n^2)$ complexity in the sequence dimension, we instead use Welford's online algorithm [46], a method that provides numerically stable variance calculations in $O(n)$ time. We gain further efficiency with a vectorized adaptation of the algorithm, allowing for GPU parallelism.

This normalization approach preserves causality and is more adaptive than a fixed per-variate scaling factor. However, in practice, we still find training instability in the presence of extreme nonstationarity. To address this, we relax our requirement of strict causality and introduce a simple clipping mechanism using variate-level statistics. We constrain $\hat{s}_t$ within a range defined by a minimum value, constant exponent $\kappa$, and the full-variate variance $s$: $\max(0.1, s \times 10^{-\kappa}) \leq \hat{s}_t \leq s \times 10^{\kappa}$, (we set $\kappa = 10$ in practice). At inference we compute statistics based solely on the historical context.

Thus, our final approach predominantly preserves causality while substantially enhancing forecasting performance, particularly for highly nonstationary series. Additional technical and implementation details are provided in Appendix A.1.

**Proportional factorized attention to judiciously capture variate interactions.** We design TOTO to natively handle multivariate forecasting by analyzing relationships in the time dimension ("time-wise" interactions) and the variate dimension ("variate-wise" interactions). While prior works that do not utilize variate-wise relationships (such as PatchTST [19] and TimesFM [12]) can still achieve competitive performance on multivariate datasets, other studies (e.g. Woo et al. [13]) have shown benefits from including variate-wise attention in ablations. However, observability metrics are often high-cardinality, multivariate time series, and a full attention schema simultaneously attending to both the time and variate dimensions (also in Woo et al. [13]) can be computationally costly.

Drawing from our experience that time relationships are often more important than cross-variate relationships, we propose a relaxation of factorized attention. Factorized attention alternates attention operations in the time and variate dimensions, allowing for time and variate mixing with lower algorithmic complexity [47–50]. While prior works typically apply both variate-wise and time-wise attention in each layer, our design provides more granular control over the relative proportion of time-wise and variate-wise interactions. Specifically, each transformer block has attention along only a single axis, and we can change the ratio of time-wise to variate-wise transformer blocks as a hyperparameter (as illustrated in Figure 2C). TOTO uses an 11:1 ratio (11 time-wise transformer blocks followed by a single variate-wise transformer block), which we found via hyperparameter optimization (see Appendix A.6).

In Section D.4, we observe that TOTO scales significantly better with increasing variates when compared to other TSFMs, consistently achieving the lowest CUDA time in its forward pass pass when the number variates is greater than 30.

**Student-T mixture model (SMM) head to model heavy-tailed observability time series.** Producing probabilistic outputs is a critical feature of time series models in several domains, including observability [51–53]. In order to produce probabilistic forecasts across the wide range of output distributions present in observability data, we employ a method based on Gaussian mixture models (GMMs), which can approximate any density fitsction [54]. We found that fitting GMMs in the presence of the extreme outliers and high skew found in observability data (see Fig. 5) leads to numerical instability in training, so we instead utilize a SMM of $K$ distributions, which robustly generalizes GMMs [55], and has shown promise for modeling heavy-tailed financial time series [56, 57]. The mixture distribution used by Woo et al. [13] contains a Student-T component alongside other distributions, in contrast to our model which solely uses Student-T components. In a contemporaneous work, Yao et al. [58] also explored time series foundation models with a Student-T mixture output. A complete mathematical formulation of our SMM, including equations and parameterizations, is provided in Appendix A.3.

**Composite robust loss to stabilize training dynamics.** Mixture models optimized via maximum likelihood are known to suffer from singularities [59] and cluster collapse [60]. We use a composite loss formulation that we find, in practice, mitigates these effects. During training, we optimize a next-patch prediction task, where the model's objective is to predict the distribution of values in the next patch given all previous patches. Our training combines the standard negative log-likelihood loss, $\mathcal{L}_{\text{NLL}}$, and a general robust loss, $\mathcal{L}_{\text{Robust}(\alpha,\delta)}$ [61] (see Appendix A.5). The robust loss provides a unified framework that allows for smoothly interpolating between several common robust loss functions [62–70], using parameters $\alpha \in [-\infty, 2]$ and $\delta > 0$ (see Fig. 7). After hyperparameter optimization, we found the Cauchy loss ($\alpha = 0$) performed best in our setting:

$$\mathcal{L}_{\text{Robust}(0,\delta)}(x_t, \hat{x}_t) = \mathcal{L}_{\text{Cauchy}}(x_t, \hat{x}_t, \delta) = \log\left(\frac{1}{2}((x_t - \hat{x}_t)/\delta)^2 + 1\right)$$

with $\delta = 0.1$. While the NLL loss utilizes the full probabilistic output of the model, the robust loss operates point-wise and measures the prediction error between the predicted SMM mean and the ground truth data point. The final combined loss used for training Toto is: $\mathcal{L} = \lambda_{\text{NLL}} \cdot \mathcal{L}_{\text{NLL}} + (1 - \lambda_{\text{NLL}}) \cdot \mathcal{L}_{\text{Robust}}$, where $\lambda_{\text{NLL}} \in [0, 1]$ is a ratio tuned simultaneously with the robust loss hyperparameters, with optimal value $\lambda_{\text{NLL}} = 0.57$. In summary, we find the composite loss critical for avoiding degenerate solutions and the robustness of the point-wise loss necessary for mitigating the effects of outliers that are common in observability data (see Section 4.3). Further details, including explicit definitions of each loss component, are provided in Appendix A.5.

### 3.2 Training data

We trained TOTO with a dataset of approximately 2.36 trillion time series points, of which 1.59 trillion are non-repeated and non-synthetic. This is significantly larger than the pretraining corpora of existing time series foundation models (Fig. 3A). Critically, 43% of our training mixture contains anonymous observability metrics from the Datadog platform. This data excludes any customer data and is sourced solely from the platform's own monitoring of internal systems. It consists only of numerical time series data, and does not include any identifiable metadata. However, much of this data is sparse, noisy, or too granular or high in cardinality to be useful in its raw form. To curate a high-quality dataset, we sample queries based on quality and relevance signals from dashboards, monitor alerts, and notebooks built by domain experts using the platform.

Alongside the Datadog data, we include public time series datasets, in particular, the GIFT-Eval Pretrain [1] and Chronos [11] collections. Importantly, we remove the subset of the Chronos datasets that overlap with the GIFT-Eval benchmark in order to avoid any leakage from the test data. We also find that adding synthetic data improves model generalization and performance. For more details on the preparation of public, synthetic, and Datadog data, please see Appendix B.

## 4 BOOM

We next introduce BOOM, a large-scale, open-source evaluation framework specifically designed to capture the unique forecasting challenges posed by modern observability workloads.

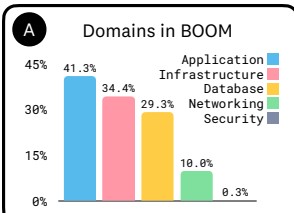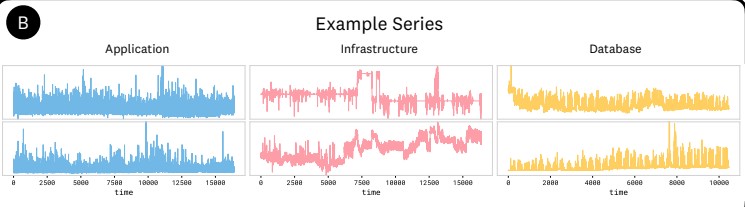

Figure 4: (A) BOOM consists of data from various domains represented within the Datadog platform. (B) Example series from three of the domains. From left to right, these series represent: sum of failed requests on a backend API, grouped by error type and source *(Application)*; CPU limits on a multi-tenant service deployed on a Kubernetes cluster, grouped by tenant *(Infrastructure)*; and sum of command executions on a Redis cache, grouped by command *(Database)*.

## 4.1 Dataset

Like the TOTO pretraining data, the observability data comprising BOOM is also sourced from the Datadog platform. However, to ensure a robust evaluation setting and prevent contamination, training data for TOTO (see Section 3.2) originates exclusively from production environments, while BOOM's evaluation data is drawn from a separate staging environment. Because isolation between production and staging data is guaranteed at the platform level, we can be certain there is no leakage. Like the training data, the BOOM data excludes any customer data and is sourced solely from the platform's own monitoring of internal systems.

BOOM's dataset consists of approximately 350 million data points across 2,807 metric queries generated through the monitoring of real-world distributed systems (Fig. 8). These series vary significantly in sampling frequency, temporal length, and cardinality, capturing realistic operational conditions (Table 1). Despite having fewer unique series compared to GIFT-Eval (2,807 vs. 144K), BOOM features substantially more total data points (350M vs. 158M) and significantly higher dimensionality, with a median of 60 variates per series compared to GIFT-Eval's predominantly univariate or low-cardinality multivariate series. Both BOOM and GIFT-Eval are significantly larger and more diverse than the older LSF benchmark. For accessibility in light of BOOM's large scale, we also provide (and evaluate on) a smaller uniformly sampled subset ("BOOMLET"), which we describe in Appendix C.3.

## 4.2 Domain taxonomy

In order to highlight the diverse characteristics BOOM's data, we assign each time series one or more labels, based on its query string, according to a taxonomy of key domains within the scope of observability monitoring. The initial labeling is conducted by prompting an LLM, which is then followed by human verification. **Application Usage** covers application interactions and user activity (e.g., request rates, API calls); **Infrastructure** includes system-level metrics (e.g., CPU usage, memory consumption); **Database** focuses on database operations and efficiency (e.g. query latency); **Networking** encompasses network monitoring signals (e.g. throughput, latency); and **Security** relates to authentication, intrusion detection, and compliance checks (e.g. log-in attempts, code vulnerabilities). We show the prevalence of each class within BOOM and provide illustrative examples in Fig. 4. Details of the taxonomy labeling procedure are provided in Appendix C.

## 4.3 Statistical characteristics

Observability time series are noted for having challenging distributional properties, including sparsity, spikes, and noisiness [71]; anomaly contamination [72] (which can manifest as skew and/or kurtosis); and for their high-cardinality multivariate nature, with complex relationships among variates [73]. They can also exhibit extreme dynamic range [31]; for example, metrics recording disk usage in bytes can range across many orders of magnitude. To illustrate these properties, we examine six relevant statistics for each variate in BOOM, GIFT-Eval, and LSF (compared in Fig. 5):

The **first-lag autocorrelation function (ACF)** measures short-term temporal dependence, with a small value indicating noisy local oscillations. Note the large lower tail in the BOOM distribution.

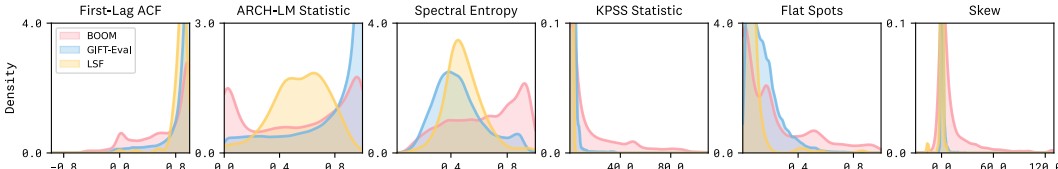

Figure 5: Distributional comparison of 6 statistical features computed on normalized time series from the BOOM GIFT-Eval, and LSF benchmark datasets. The broader and shifted distributions in the BOOM series reflect the increased diversity, irregularity, and nonstationarity characteristic of observability data.

The **ARCH-LM statistic** [74] detects autoregressive conditional heteroskedasticity. Lower values suggest time-varying volatility; the BOOM distribution is bimodal with a large peak near zero.

**Spectral entropy** [73], which quantifies the unpredictability of a series in the frequency domain, is also higher on average in observability series. This indicates less periodicity and greater irregularity.

The **KPSS statistic** [75], a test of nonstationarity, and assumes larger values in the BOOM observability data, suggesting more frequent deviations from a deterministic trend.

**"Flat spots"** [76] measures the longest constant subsequence with a time series, normalized by series length. This metric is higher in BOOM, indicative of sparse metrics within observability data.

**Skew** reveals heavier-tailed, asymmetric distributions in observability data, often reflecting bursty behavior and rare but extreme events.

Overall, we find that the more extreme values of each of these statistics in the BOOM data, reinforces the distinct and challenging nature of observability times series (see also Fig. 1B).

## 4.4 Evaluation protocol

To evaluate models on BOOM we closely follow the evaluation methodology proposed by Aksu et al. [1] for GIFT-Eval, including its standardized prediction lengths, strides, and train/validation/test splits. As in GIFT-Eval, our key accuracy metrics are Mean Absolute Scaled Error (MASE) [77] and approximate Continuous Ranked Probability Score (CRPS) [78, 79], both normalized by the Seasonal Naive forecast. And like GIFT-Eval, we also compute an average rank (where rank is computed as the mean rank with respect to CRPS across all prediction tasks).

We deviate from the GIFT-Eval protocol in two key respects in order to handle the presence of metrics with mostly constant values and infrequent spikes, which can cause problems in computing traditional metrics. First, we employ a shifted geometric mean for aggregating MASE and CRPS across tasks, which is stable in scenarios where constant subsequences would otherwise cause a standard geometric mean to collapse [80, 81]. Second, we use a separate evaluation approach for a small subset of the time series in BOOM where extreme cases of constant subsequences still cause numerical instability for naive-forecast-normalized metrics. See Appendix C for details.

| Dataset | # Series | # Variates | # Points (M) | Interval | | Series Length | | | # Variates Per Series | | | Pred. Length | |
|---|---|---|---|---|---|---|---|---|---|---|---|---|---|
| | | | | Min | Max | Min | Med | Max | Min | Med | Max | Min | Max |
| BOOM | 2,807 | 32,887 | 350 | 10 sec. | 1 day | 101 | 13,631 | 16,384 | 1 | 60 | 100 | 48 | 900 |
| BOOMLET | 32 | 1,627 | 23 | 10 sec. | 1 hr. | 5,231 | 16,384 | 16,384 | 21 | 49 | 100 | 48 | 900 |
| GIFT-Eval | 144,246 | 147,688 | 158 | 10 sec. | 1 yr. | 19 | 1,043 | 140,256 | 1 | 1 | 21 | 6 | 900 |
| LSF | 6 | 370 | 11 | 10 min. | 1 hr. | 17,420 | 39,500 | 69,680 | 7 | 7 | 321 | 96 | 720 |

Table 1: Key statistics of the time series benchmarks used for evaluation in this study. BOOM, introduced in this paper, contains approximately twice as many total time series points as GIFT-Eval (350M vs. 158M) and vastly more than LSF. BOOM, consists of multivariate series of much higher average cardinality than the other benchmarks, with a median of 60 variates per series vs. only 1 for GIFT-Eval and 7 for LSF. Conversely, BOOM contains a narrower range of intervals compared to GIFT-Eval, ranging from 10 seconds to 1 day, vs. 10 seconds to 1 year for GIFT-Eval. This reflects the generally shorter time scales relevant to the observability domain. BOOMLET, a smaller subset of BOOM that we provide for easier evaluation, exhibits similar statistics.

# 5 Experiments

We evaluate TOTO on three benchmarks: BOOM, GIFT-Eval, and LSF (see Table 1). We compare against a comprehensive set of methods, including zero-shot foundation models ('Zero Shot'), neural models trained on the target data ('Full Shot'), and classical supervised approaches ('Baselines').

**BOOM**. We evaluate TOTO's zero-shot forecasting performance alongside other foundation models, [11–14, 23, 24], as well as full-shot statistical baselines. We do not evaluate full-shot deep learning models on BOOM; as discussed in Section 1, these models are often impractical in real-word observability systems at scale. Similarly, we were unable to evaluate TabPFN [82] (a recent zero-shot model with strong performance on GIFT-Eval) on BOOM, as its open-source implementation's lack of support for batched inference made it impractically slow to run. Details of the inference settings and evaluation procedures for all models are described in Appendix C.2. As shown in Table 2, TOTO consistently outperforms other models, achieving 13.1% and 12.4% lower MASE and CRPS, respectively, than the next best (Moirai$_{Base}$), and a significantly lower rank (2.351 vs. 4.278). Similar qualitative results hold on the BOOMLET subset (Appendix D.1.1).

| Dataset | Metric | Zero Shot | | | | | | | Baselines | | | | |
|---|---|---|---|---|---|---|---|---|---|---|---|---|---|
| | | TOTO | Moirai$_{Base}$ | TimesFM$_{2.0}$ | Chronos$_{Bolt-Base}$ | Timer | Time-MoE$_{Base}$ | VisionTS | Auto-ARIMA | Auto-ETS | Auto-Theta | DLinear | DeepAR |
| BOOM | MASE ↓ | **0.617** | 0.710 | 0.725 | 0.726 | 0.796 | 0.881 | 0.988 | 0.824 | 0.842 | 1.123 | - | - |
| | CRPS ↓ | **0.375** | 0.428 | 0.447 | 0.451 | 0.639 | 0.643 | 0.673 | 0.736 | 1.975 | 1.018 | - | - |
| | Rank ↓ | **2.369** | 4.328 | 5.243 | 5.576 | 9.920 | 9.843 | 10.989 | 9.686 | 11.671 | 12.472 | - | - |
| BOOMLET | MASE ↓ | **0.617** | 0.779 | 0.685 | 0.711 | 0.807 | 0.793 | 0.912 | 0.922 | 0.969 | 1.030 | 0.823 | 0.883 |
| | CRPS ↓ | **0.519** | 0.630 | 0.603 | 0.637 | 0.793 | 0.780 | 0.885 | 0.880 | 15.664 | 1.182 | 0.641 | 0.697 |
| | Rank ↓ | **1.300** | 5.244 | 4.844 | 6.411 | 11.344 | 11.011 | 13.278 | 11.778 | 14.156 | 14.522 | 6.044 | 7.822 |

Table 2: **BOOM results.** Performance of TOTO, other zero-shot models, and baselines. MASE and CRPS are normalized by the Seasonal Naive forecast and aggregated across tasks using shifted geometric mean. Rank is the mean rank across tasks with respect to CRPS. For model families with multiple sizes (Moirai, Chronos) we show the best-performing variant. TOTO significantly outperforms other methods on all metrics. We observe similar qualitative trends on the BOOMLET results. Additional results, including all model sizes evaluated as well as categorical breakdowns, are available in Appendix D.1. Key: **Best results**, Second-best results.

**GIFT-Eval**. We evaluate TOTO's zero-shot performance on general-purpose time series forecasting via the GIFT-Eval benchmark [1]. TOTO achieves the top performance among all reported models, with an average ranking score of 5.495 as of May 2025. It achieves strong results both in point forecasting, with a MASE of 0.673, and probabilistic forecasting, with a CRPS of 0.437. Details of our evaluation settings for GIFT-Eval are provided in Appendix D.2. Notably, TOTO is the top-performing method in spite of the fact that several competing models have known partial data leakage with the benchmark, as discussed by Aksu et al. [1].

| Metric | Zero Shot | | | | | Full Shot | | | | Baselines | | |
|---|---|---|---|---|---|---|---|---|---|---|---|---|
| | TOTO | Moirai$_{Large}$ | TimesFM$_{2.0}$ | Chronos$_{Bolt-Base}$ | TabPFN-TS | TEMPO | TTM-R2 | PatchTST | TFT | Auto-ARIMA | Auto-ETS | Auto-Theta |
| MASE ↓ | **0.673** | 0.785 | 0.680 | 0.725 | 0.748 | 0.773 | 0.679 | 0.762 | 0.822 | 0.964 | 1.088 | 0.978 |
| CRPS ↓ | 0.437 | 0.506 | 0.465 | 0.485 | 0.480 | **0.434** | 0.492 | 0.496 | 0.511 | 0.770 | 6.327 | 1.051 |
| Rank ↓ | **5.495** | 10.330 | 8.412 | 8.309 | 8.402 | 8.897 | 10.103 | 10.268 | 11.629 | 21.608 | 25.134 | 24.134 |

Table 3: **GIFT-Eval results.** TOTO's performance vs. the top Zero Shot, Full Shot, and Baseline models reported on the leaderboard [3]. MASE and CRPS are normalized by the Seasonal Naive forecast and aggregated across tasks using geometric mean. Rank is the mean rank across tasks with respect to CRPS. For multi-size models (Moirai, Chronos) we show the best-performing variant. TOTO achieves top performance on GIFT-Eval in Rank and MASE, and second-best performance in CRPS. Key: **Best results**, Second-best results.

**LSF**. We evaluate TOTO on the widely-used Long Sequence Forecasting (LSF) benchmark [2]. Despite known limitations regarding dataset diversity and saturation by supervised methods [1, 29], TOTO achieves state-of-the-art results in zero-shot evaluations, attaining the best performance on 8 out of 12 reported metrics when compared against other zero-shot methods, and the lowest average MAE and MSE (see Table 4). Furthermore, while our focus in this work is on the zero-shot setting, we explored the efficacy of fine-tuning TOTO on the training splits of LSF and report the results in Table 15. We find that TOTO achieves state-of-the-art results in full-shot evaluations, also attaining the best performance on 8 out of 12 reported metrics, and the lowest average MAE and MSE of all methods. These results underscore TOTO's strong generalization capabilities and its effectiveness when fine-tuned on specialized datasets, making it a versatile choice for a wide range of time series forecasting tasks. See Appendix D.3 for full results and additional experimental details.

| Dataset | Metric | Toto | Moirai$_{Small}$ | Moirai$_{Base}$ | Moirai$_{Large}$ | VisionTS | Time-MoE$_{Base}$ | Time-MoE$_{Large}$ | Time-MoE$_{Ultra}$ |
|---|---|---|---|---|---|---|---|---|---|
| | | | | | | *Zero Shot* | | | |
| ETTh1 | MAE ↓ | **0.413** | 0.424 | 0.438 | 0.469 | 0.414 | 0.424 | 0.419 | 0.426 |
| | MSE ↓ | 0.435 | 0.400 | 0.434 | 0.510 | **0.390** | 0.400 | 0.394 | 0.412 |
| ETTh2 | MAE ↓ | **0.363** | 0.379 | 0.382 | 0.376 | 0.375 | 0.404 | 0.415 | 0.399 |
| | MSE ↓ | 0.340 | 0.341 | 0.345 | 0.354 | **0.333** | 0.366 | 0.405 | 0.371 |
| ETTm1 | MAE ↓ | 0.378 | 0.409 | 0.388 | 0.389 | **0.372** | 0.415 | 0.405 | 0.391 |
| | MSE ↓ | 0.396 | 0.448 | 0.381 | 0.390 | 0.374 | 0.394 | 0.376 | **0.356** |
| ETTm2 | MAE ↓ | **0.303** | 0.341 | 0.321 | 0.320 | 0.321 | 0.365 | 0.361 | 0.344 |
| | MSE ↓ | **0.267** | 0.300 | 0.272 | 0.276 | 0.282 | 0.317 | 0.316 | 0.288 |
| Electricity | MAE ↓ | **0.243** | 0.320 | 0.274 | 0.273 | 0.294 | - | - | - |
| | MSE ↓ | **0.161** | 0.233 | 0.188 | 0.188 | 0.207 | - | - | - |
| Weather | MAE ↓ | **0.245** | 0.267 | 0.261 | 0.275 | 0.292 | 0.297 | 0.300 | 0.288 |
| | MSE ↓ | **0.224** | 0.242 | 0.238 | 0.259 | 0.269 | 0.265 | 0.270 | 0.256 |
| Mean | MAE ↓ | **0.324** | 0.357 | 0.344 | 0.350 | 0.345 | - | - | - |
| | MSE ↓ | **0.304** | 0.327 | 0.310 | 0.330 | 0.309 | - | - | - |
| Best Count | | **8** | **0** | **0** | **0** | **3** | **0** | **0** | **1** |

Table 4: **LSF results** Zero-Shot comparison of models on the LSF benchmark. Non-TOTO values are reproduced from published tables. Key: **Best results**, Second-best results. "Best Count" row reports the number of times each model attains the best result for a given dataset-metric pair.

# 6    Conclusion

This work reframes time series forecasting through the lens of observability—a domain marked by scale, complexity, and real-world urgency. We presented TOTO, a foundation model purpose-built to forecast multivariate observability metrics with zero-shot accuracy, and introduced BOOM, the first benchmark to capture the messy, high-cardinality reality of production telemetry data at scale. TOTO advances the frontier in zero-shot time series forecasting and sets new state-of-the-art results on BOOM, GIFT-Eval, and LSF. A limitation of TOTO (and other TSFMs) is the assumption of fixed-interval times series. Currently, we address missing points with heuristic imputations; natively handling missing data may provide a better solution. Additionally, TOTO does not directly incorporate calendar-based features—an interesting direction for future work. Finally, a study of performance at extreme prediction lengths would be insightful for certain applications. By open-sourcing both model and benchmark, we hope to accelerate research to answer these and other open questions, contribute to the community, and to draw attention to an important real-world application.

# 7    Acknowledgements

Our work is made possible by the efforts of numerous teams at Datadog and beyond. Special thanks and acknowledgment to:

Aaron Taa, Alexis Lê-Quôc, Amine Naouas, Antoine Gaillard, Ara Pulido, August Marsalis, Ben Donohue, Ben Hinthorne , Benedetto Buratti, Bharath Vontimitta, Bill Birkholz, Brendan Rhoads, Charuprabha Gaur, Damián Vicino, Dan Haggerty, Dengke Liu, Dimitrios Gklezakos, Dominique West, Emilie Xu, Erica Hale, Gerald Woo, Jake Femminineo, Jake Hooker, Janine Kromhout, Jared Ledvina, Jared Schifrien, Jeremy Garcia, Jeromy Carriere, Jesse Mack, Jessica Cordonnier, Joe Jones, Johan Andersen, Kathy Nguyen, Kevin Beach, Luca Pizzamiglio, Madison Moore, Marion Chan-Renous, Max Livingston, Maxim Brown, Maxime Visonneau, Mayeul Blanzat, Michael Hoang, Mikhail Khodak, Mononito Goswami, Nick Sollecito, Olivier Pomel, Phil Sarin, Quentin François, Quentin Gendre, Raya Wakil, Rikki Endsley, Roashan Ayene, Rob Boll, Romoli Bakshi, Sajid Mehmood, Sean O'Connor, Steven Zhou, Vyom Shah, and Zakaria Fikrat.

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

# A Model architecture details

## A.1 Input/output scaling

To ensure numerical stability, we compute the $\hat{\mu}_t$ and $\hat{s}_t$ using an efficient vectorized implementation (Listing 1) of Welford's online algorithm [46], incorporating Bessel's correction to provide an unbiased estimator of variance, as described in Option A of [83]. We stabilize training against extreme outliers by incorporating weak information from the global statistics.

```python
def compute_causal_statistics(
    data: torch.Tensor,
    weights: torch.Tensor,
    minimum_scale: float,
) -> Tuple[torch.Tensor, torch.Tensor]:
    # Compute causal means at each time step
    weighted_data = weights * data
    cum_weights = torch.cumsum(weights, dim=-1)
    cum_values = torch.cumsum(weighted_data, dim=-1)
    denominator = cum_weights.clamp_min(1.0)
    causal_means = cum_values / denominator

    # For Welford's algorithm, we need to compute the correction term
    # delta using the difference between the current value and the
    # previous running mean.
    shifted_means = torch.zeros_like(causal_means)
    shifted_means[..., 1:] = causal_means[..., :-1]
    delta = data - shifted_means

    # Compute m_2, the second moment accumulator for Welford's
    # algorithm.
    increment = delta * (data - causal_means) * weights
    m_2 = torch.cumsum(increment, dim=-1)

    # Compute the variance using Bessel's correction.
    causal_variance = m_2 / torch.clamp(denominator - 1.0, min=1.0)
    causal_scale = torch.sqrt(causal_variance + minimum_scale)

    return causal_means, causal_scale
```

Listing 1: Vectorized PyTorch implementation of Welford's algorithm for computing causal statistics

In our ablation study (Section E), we find that causal scaling leads to dramatic performance improvements over naive global scaling.

## A.2 Attention mechanism

To address the unique challenges of time series data, and particularly to adapt transformer architectures for multivariate time-series forecasting, several works have implemented modifications to the attention mechanism. These strategies have included:

- Concatenating variates along the time dimension and computing full self-attention between every variate/time location, as in the "any-variate attention" used by Woo et al. [13]. This can capture every possible variate and time interaction, but it is costly in terms of computation and memory usage.

- Assuming variate independence, and computing attention only in the time dimension as in Nie et al. [19], Shi et al. [14]. This is efficient, but throws away all information about variate-wise interactions.

- Computing attention only in the variate dimension, and using a feed-forward network in the time dimension [84, 35].

- Computing "factorized attention," where each transformer block contains a separate variate and time attention computation [47–49]. This allows both variate and time mixing, and is

more efficient than full cross-attention. However, it doubles the effective depth of the network.

In Section 3.1, we propose a novel approach that allows for both variate and time interactions, while reducing the computational cost and improving overall scalability.

### A.2.1 Complexity analysis

After the patchwise embedding layer, we have inputs of shape $\mathbf{X} \in \mathbb{R}^{B \times M \times \frac{L}{P} \times D}$, where $B$ is the batch dimension, $M$ is the number of variates per batch item, $\frac{L}{P}$ is time steps divided by patch width, and $D$ is the model embedding dimension.

**Time-wise attention.** We parallelize along the time dimension by reshaping the input tensor from 4 dimensions to 3:

$$\mathbf{X} \in \mathbb{R}^{B \times M \times \frac{L}{P} \times D} \to \mathbf{X}_{\text{time}} \in \mathbb{R}^{(B \times M) \times \frac{L}{P} \times D}$$

This allows for attention to be calculated independently in parallel per variate, giving a complexity of:

$$\mathcal{O}(M \times (\frac{L}{P})^2 \times D)$$

In the time-wise attention blocks, we use causal masking and rotary positional embeddings [43] with XPOS [44] in order to autoregressively model time-dependent features.

**Variate-wise attention.** We similarly parallelize along the variate dimension by reshaping the input tensor:

$$\mathbf{X} \in \mathbb{R}^{B \times M \times \frac{L}{P} \times D} \to \mathbf{X}_{\text{variate}} \in \mathbb{R}^{(B \times \frac{L}{P}) \times M \times D}$$

We calculate attention in parallel for each time step, with complexity:

$$\mathcal{O}(\frac{L}{P} \times M^2 \times D)$$

In the variate-wise blocks, we use full bidirectional attention (without causal masking) in order to preserve permutation invariance of the covariates, with a block-diagonal ID mask to ensure that only related variates attend to each other. This masking allows us to pack multiple independent multivariate time series into the same batch, in order to improve training efficiency and reduce the amount of padding.

**Computational complexity.** Each transformer block in our model contains $N$ time-wise attention layers and 1 variate-wise layer. The complexity for full self-attention over $N + 1$ layers, where interactions can occur across all variates and sequence positions, would be of complexity:

$$\mathcal{O}\left((N+1) \times M^2 \times \left(\frac{L}{P}\right)^2 \times D\right) \tag{1}$$

This reflects the quadratic dependence on both the sequence length $\frac{L}{P}$ and the variate dimension $M$, with linear dependence on the embedding dimension $D$. However, by utilizing factorized attention, we can reduce the computational complexity of the attention calculation to:

$$\mathcal{O}\left(N \times M \times \left(\frac{L}{P}\right)^2 \times D + \frac{L}{P} \times M^2 \times D\right) = \\ \mathcal{O}\left(D \times \frac{L}{P} \times M \times \left(N \times \frac{L}{P} + M\right)\right) \tag{2}$$

We demonstrate that factorized variate-wise attention is asymptotically smaller in computational complexity than full self-attention (see Equation 1 and Equation 2). When comparing a model with full self-attention, we can assume $N$ and $D$ are fixed. Therefore:

$$\mathcal{O}\left( M \times \left( \frac{L}{P} \right)^2 + \frac{L}{P} \times M^2 \right) < \mathcal{O}\left( M^2 \times \left( \frac{L}{P} \right)^2 \right)$$

which reduces to:

$$\mathcal{O}\left( M + \frac{L}{P} \right) < \mathcal{O}\left( M \times \frac{L}{P} \right).$$

Thus, by factorizing attention into time-wise and variate-wise components, the computational complexity is reduced, especially for large numbers of variates $M$ or long sequences $\frac{L}{P}$, making it more scalable than full self-attention.

### A.3 Probabilistic prediction

Practitioners who rely on time series forecasting typically prefer probabilistic predictions. A common practice in neural time series models is to use an output layer where the model regresses the parameters of a probability distribution. This allows for prediction intervals to be computed using Monte Carlo sampling (see Appendix A.4) [85].

Common choices for an output layer are Normal [85] and Student-T [86, 21], which can improve robustness to outliers. Moirai [13] allows for more flexible residual distributions by proposing a novel mixture model incorporating a weighted combination of Gaussian, Student-T, Log-Normal, and Negative-Binomial outputs.

However, real-world time series can often have complex distributions that are challenging to fit, with outliers, heavy tails, extreme skew, and multimodality. In order to accommodate these scenarios, we introduce an even more flexible output likelihood in Section 3.1 based on a SMM [55].

TOTO makes predictions using a mixture of $K$ Student-T distributions (where $K$ is a hyperparameter) for each time step, as well as a learned weighting. Formally, the SMM is defined by:

$$p(x) = \sum_{k=1}^{K} \pi_k \mathcal{T}(x \mid \mu_k, \tau_k, \nu_k) \tag{3}$$

where $\pi_{k \in K}$ are nonnegative mixing coefficients which sum to 1 for the $k$th Student's t-distribution $\mathcal{T}_k$ with $\nu_k$ degrees of freedom, mean $\mu_k$, and scale $\tau_k$. $\mathcal{T}(x \mid \mu, \sigma, \nu)$ is defined as:

$$\mathcal{T}(x \mid \mu, \tau, \nu) = \frac{\Gamma\left( \frac{\nu+d}{2} \right)}{\Gamma\left( \frac{\nu}{2} \right) (\nu\pi)^{d/2} |\tau|^{1/2}} \left( 1 + \frac{1}{\nu}(x-\mu)^\top \tau^{-1}(x-\mu) \right)^{-\frac{\nu+d}{2}}, \tag{4}$$

where $\Gamma(\cdot)$ is the gamma function.

In our ablation study (Appendix E), we find that the SMM improves both point prediction and probabilistic forecasting accuracy when compared with a single Student-T distribution as used in TiDE [86], Lag-Llama [21], and implementations of DeepAR [85], PatchTST [19], iTransformer [42], and others in the popular open-source GluonTS library [87].

The parameters of this mixture model are computed from the flattened features $h_t \in \mathbb{R}^D$ produced by the transformer backbone for each time step $t$, where $D$ is the model's embedding dimension. Using a set of linear projections with weight matrices $W \in \mathbb{R}^{K \times D}$ and bias vectors $b \in \mathbb{R}^K$, we derive all $K$ mixture components simultaneously. For each time step $t$, the parameters are computed

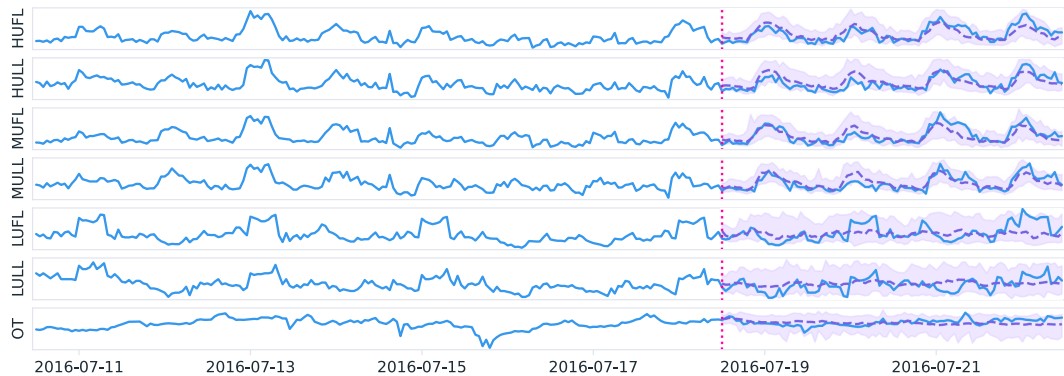

Figure 6: Example of TOTO 's 96-step zero-shot forecasts on the ETTh1 dataset, showing multivariate probabilistic predictions. Solid lines represent ground truth, dashed lines represent median point forecasts, and shaded regions represent 95% prediction intervals.

as:

$$\nu_t = 2 + \max(\text{softplus}(W_\nu h_t + b_\nu), \epsilon) \tag{5}$$

$$\mu_t = W_\mu h_t + b_\mu \tag{6}$$

$$\tau_t = \max(\text{softplus}(W_\tau h_t + b_\tau), \epsilon) \tag{7}$$

$$\tilde{\pi}_t = W_\pi h_t + b_\pi \tag{8}$$

where each equation produces a vector in $\mathbb{R}^K$ containing the parameters for all mixture components at time $t$. The individual component parameters $\nu_{t,k}$, $\mu_{t,k}$, $\tau_{t,k}$, and $\tilde{\pi}_{t,k}$ (the mixture logits) are the $k$th elements of these vectors. The parameter $\epsilon$ is a small positive constant, and softplus($x$) = $\log(1 + e^x)$. The use of softplus and $\epsilon$ ensure that the scale $\tau$ remains positive. Similarly, we add the constraint $\nu > 2$ to ensure that each component of our mixture has well-defined first and second moments (mean and variance).

The mixture weights $\pi$ are computed using by applying softmax to the logits:

$$\pi_{t,k} = \text{softmax}(\tilde{\pi}_t, k) = \frac{e^{\tilde{\pi}_{t,k}}}{\sum_{j=1}^{K} e^{\tilde{\pi}_{t,j}}} \tag{9}$$

An example distribution median and 95th percentile is illustrated in Fig. 6.

### A.4 Forecasting

When performing inference, we draw $u$ (for some user specified integer $u > 0$) samples from the mixture distribution at each timestamp, then feed each sample back into the decoder for the next prediction, resulting in $n$ identically and independently sampled time-series. This allows us to produce prediction intervals at any quantile, limited only by the number of samples. Our exact sampling procedure for several tasks is detailed in Section C.2.

### A.5 Loss function

TOTO learns the conditional distribution $p(X_{i+1}|X_{1:i})$, where $X_i$ represents the $i$-th patch containing multiple time steps.

The $\mathcal{L}_{\text{NLL}}$ optimizes probabilistic predictions and is defined as:

$$\mathcal{L}_{\text{NLL}}(x, \mu, \tau, \nu) = -\log\left(p(x_t|X_{1:i})\right) = -\log\left(\sum_{k=1}^{K} \pi_{t,k} \mathcal{T}(x_t \mid \mu_{t,k}, \tau_{t,k}, \nu_{t,k})\right) \tag{10}$$

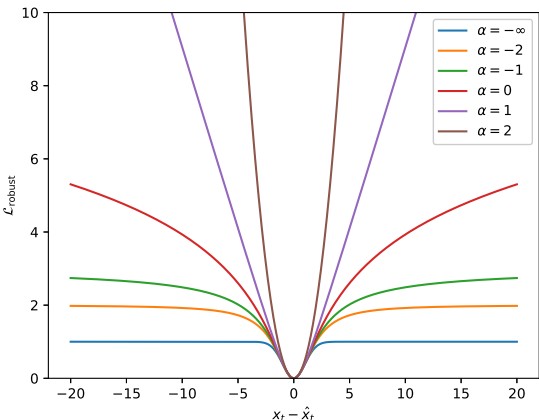

Figure 7: Visualization of generalized robust loss for different values of $\alpha$, with $\delta$ fixed at 1. Changing $\delta$ scales the horizontal axis.

where $p(x_t|X_{1:i})$ is the probability density of the ground truth $x_t$ under the model's predicted mixture distribution conditioned on all previous patches. The parameters $\pi_{t,k}$, $\mu_{t,k}$, $\tau_{t,k}$, and $\nu_{t,k}$ are the mixture weights and Student-T parameters computed by the model for time step $t$.

For a ground truth value $x_t$ in patch $i+1$ and the mean prediction $\hat{x}_t = \mathbb{E}[p(x_t|X_{1:i})]$, the robust loss is defined below [61]:

$$
\mathcal{L}_{\text{Robust}(\alpha,\delta)}(x_t, \hat{x}_t) = \begin{cases} \frac{1}{2}((x_t - \hat{x}_t)/\delta)^2, & \alpha = 2 \\ \log\left(\frac{1}{2}((x_t - \hat{x}_t)/\delta)^2 + 1\right), & \alpha = 0 \\ 1 - \exp\left(-\frac{1}{2}((x_t - \hat{x}_t)/\delta)^2\right), & \alpha = -\infty \\ \frac{|\alpha-2|}{\alpha}\left[\left(\frac{((x_t - \hat{x}_t)/\delta)^2}{|\alpha-2|} + 1\right)^{\alpha/2} - 1\right], & \text{otherwise} \end{cases}
\tag{11}
$$

Here, $\mathcal{L}_{\text{Robust}(\alpha,\delta)}$ serves as a point prediction error measure, where $\alpha \leq 2$ is a shape parameter that controls the robustness to outlier observations (Fig. 7) and $\delta > 0$ is a scale parameter that determines the size of the parabolic portion of the loss curve. This loss component directly penalizes point prediction accuracy, and we conjecture this may help steer the mixture model away from degenerate solutions of the type described in [60]. In our ablation study, we find that adding the robust loss component significantly improves point forecasting accuracy without hurting probabilistic predictions (Section E).

$\mathcal{L}$ is applied to each timestep $t$ in the target patch $X_{i+1}$, and the total loss is aggregated across all timesteps during training. By combining the probabilistic $\mathcal{L}_{NLL}$ loss with the robust point-prediction loss, we achieve both accurate distribution modeling and stable convergence, especially in domains with highly heterogeneous data characteristics. The hyperparameter $\lambda_{\text{NLL}}$ controls the balance between these two loss components and is tuned empirically.

## A.6 Hyperparameter optimization

To determine the optimal architecture and training configuration for Toto, we conducted an extensive hyperparameter sweep using Optuna [88], a Bayesian optimization framework. We employed the Tree-structured Parzen Estimator (TPE) algorithm to efficiently explore the high-dimensional search space.

Our optimization objective was to minimize the validation mean absolute error (MAE) on multi-step forecasting tasks on a random validation split of the observability portion of the pretraining data. We train the model using the AdamW optimizer [89] with a WSD learning rate scheduler [90]. We performed this sweep for 133 iterations over ranges described in Table 5, each for 50,000 training steps.

| Category | Values / Ranges |
|---|---|
| Patch Size | $\{16, 32, 64\}$ |
| Variate-wise Attention Frequency | Every $\{3, 4, 6, 12\}$ layers |
| Variate-wise Layer First | [True, False] |
| $\mathcal{T}$ Components | [8, 16, 24, 32] |
| Loss Function | $\lambda_{\text{NLL}} \in [0.05, 1.0]$ |
| Robust Loss Params | $\alpha \in \{-\infty, -2, 0, 0.5, 1.0\}, \delta \in [0.1, 3.0]$ |
| Warmup Steps | [0, 10,000] |
| Stable Ratio* | [.1, .9] |
| Learning Rate | $[10^{-5}, 5 \times 10^{-3}]$ |
| Weight Decay | $[10^{-3}, 10^{-1}]$ |
| Synthetic Data Proportion | [0.0, 0.75] |
| Shuffling Type | [Normally Distributed, Adjacent, Random, None] |
| Normally Distributed Shuffling Standard Deviation | [.15, 5000] |
| Shuffling Frequency | [0.0, 0.3] |

Table 5: Summary of hyperparameter search space. *Stable Ratio defines the proportion of steps that are stable after the warmup phase of the WSD learning rate schedule.

The resulting hyperparameter configuration described in Table 6 obtained the best multistep (average of 96 and 192) MAE on the Datadog validation set.

| Hyperparameter | Value |
|---|---|
| Embedding Dimension | 768 |
| MLP Dimension | 3072 |
| # Layers | 12 |
| # Heads | 12 |
| # Variates | 32 |
| Spacewise Layer Cadence | 12 |
| Patch Size | 64 |
| # $\mathcal{T}$ Mixture Model Components | 24 |
| Annealing Schedule | WSD |
| Optimizer | AdamW |
| $(\beta_1, \quad \beta_2)$ | (0.9579, 0.9581) |
| Weight Decay | 0.0014 |
| Initial Learning Rate | 0.0005 |
| Warmup Steps | 6784 |
| Stable Steps | 112,255 |
| Decay Steps | 15,962 |
| Batch Size | 128 |
| Total Train Steps | 135,001 |
| $\mathcal{L}_{\text{Robust}} \alpha$ | 0.0000 |
| $\mathcal{L}_{\text{Robust}} \delta$ | 0.1010 |
| $\lambda_{\text{NLL}}$ | 0.5755 |
| $\kappa$ | 10 |

Table 6: Hyperparameters for Toto

In Section E, we perform an ablation study on the impact of various model components. We optimize speed and memory usage by utilizing fused kernel implementations and memory efficient attention operations via xformers [91], (with the FlashAttention-3 kernel [92]).

We ran all experiments, including hyperparameter tuning, final model training, and benchmark evaluation on a GPU cluster consisting of A100s and H100s.

# B Training data preprocessing

## B.1 Observability dataset

Observability metrics are retrieved from a large-scale time series database using a specialized query language supporting filters, group-bys, time aggregation, and various transformations and postprocessing functions (Fig. 8). We consider groups returned from the same query to be related variates in a multivariate time series. After we retrieve the query results, we discard the query strings and group identifiers, keeping only the raw numeric data. As described in Section 3.2, we source metrics defined by user-generated queries. This excludes any customer data and is sourced solely from the internal users and telemetry.

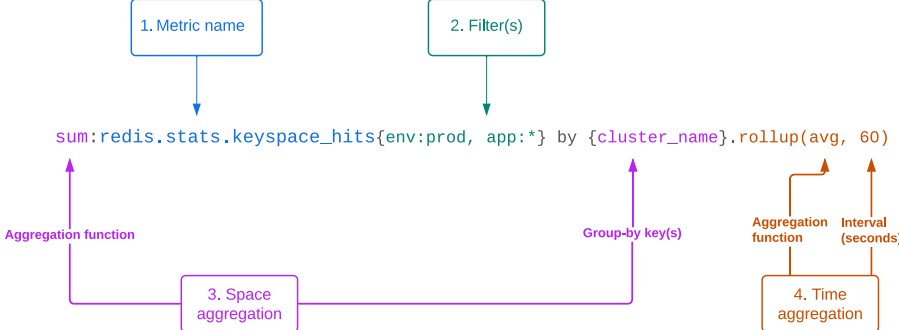

Figure 8: Example metric query in the Datadog platform. The metric name (1) determines which metric is being queried. The filter clause (2) limits which contexts are queried, in this case restricting the query to apps in the prod environment. The space aggregation (3) indicates that the sum of the metric value should be returned for each unique value of the group-by key(s), aggregated across all other keys. The time aggregation (4) indicates that metric values should be aggregated to the average for each 60-second interval. The query results will be a multivariate time series with 1-minute time steps, and with separate individual variates for each unique value of cluster_name.

## B.2 Public datasets

We train on a public dataset corpus, which exposes the model to diverse time series behaviors across different domains and sampling frequencies. Our pre-training dataset incorporates a diverse collection of time series from the GIFT-Eval Pretrain collection [26] and non-overlapping Chronos datasets [11]. These datasets include `ercot`, `exchange_rate`, `weatherbench_daily`, `weatherbench_hourly`, `weatherbench_monthly`, `dominick`, `mexico_city_bikes`, `ushcn_daily`, and `wiki_daily_100k`.

## B.3 Synthetic data

We supplement our training with synthetic data to further improve model performance. Our synthetic dataset consists of procedurally generated time series using an approach similar to TimesFM [12], as well as `kernel_synth_1m` from the Chronos dataset [11]. Synthetic data constitutes approximately 33% of our training dataset.

We generate synthetic time series through the composition of components such as piecewise linear trends, ARMA processes, sinusoidal seasonal patterns, and various residual distributions. Our procedural generation randomly combines multiple processes per variate to introduce diverse patterns. The generation includes creating base series with transformations, clipping extreme values, and rescaling to specified ranges.

These synthetic datasets help the model learn robust representations by providing examples with specific characteristics that might be underrepresented in real-world data.

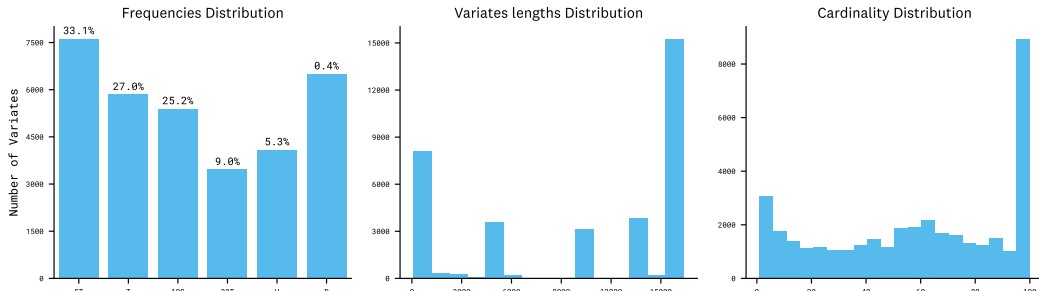

Figure 9: Representative figure showing the metadata breakdown by variate in the dataset: (left) sampling frequency distribution – bar heights show the number of *variates* with each frequency, while the percentages show the fraction of overall *observations*, (middle) series length distribution, and (right) number of variates distribution.

## B.4 Preprocessing

To prepare the raw time series for training, we apply padding and masking techniques to align the series lengths, making them divisible by the patch stride. This involves adding necessary left-padding to both the time series data and the ID mask, ensuring compatibility with the model's requirements.

Next, various data augmentations are employed to enhance the dataset's robustness. We introduce random time offsets to prevent memorization caused by having series always align the same way with the patch grid. After concatenating the Datadog and public datasets for training, we also implement a variate shuffling strategy to maintain diversity and representation. Specifically, we randomly combine variates from either Datadog, open source datasets (GIFT-Eval pretrain and Chronos datasets), and/or synthetic data with a probability of 14%, thus creating new, diverse combinations of data points. We shuffle series with adjacent indices (batched by 32 variates), favoring data points that were closer together in the original datasets. This approach improves the model's ability to generalize across different types of data effectively.

## C  BOOM

### C.1  Domain taxonomy

BOOM data is collected using the same query language as described in Section B.1. Each metric query yields a collection of time series—one per unique attribute combination—resulting in multivariate time series with attributes serving as variates. The distribution of aggregation metrics collected is described in Table 7. In Table 8, we categorize the series into 5 major groups based on the domain described within the query. As part of the labeling process, a large language model was used to pre-fill labels from the metric names, which were then manually reviewed by human annotators to ensure consistency and accuracy.

| Metric Type | Description | Proportion (%) |
|---|---|---|
| **Gauge** | Last measurement reported within intervals | 65.7 |
| **Rate** | Number of event occurrences per second | 26.8 |
| **Distribution** | Aggregated statistical summaries across sources (e.g., average, percentiles) | 5.3 |
| **Count** | Total number of event occurrences within intervals | 2.2 |

Table 7: Taxonomy of metric types in the benchmark dataset with their relative proportions.

Each time series in the BOOM undergoes a standardized preprocessing pipeline designed to address the noise, irregularities, and high cardinality typical of observability data. First, missing intervals, common in telemetry due to irregular metric emission, are filled using metric-aware strategies: count-based metrics are zero-filled under the assumption that missing values reflect inactivity, while real-valued metrics (e.g., rates or gauges) are linearly interpolated. Following imputation, series are sliced into fixed-length windows of up to 16,384 points. Unlike the training set, which uses random offsets and padding to augment diversity, the benchmark slices are extracted without offset or

| System Domain | Description | Proportion (%) |
|---|---|---|
| **Application Usage** | Covers application interactions and user activity (e.g., request rates, API calls) | 41.3 |
| **Infrastructure** | System-level metrics (e.g., CPU usage, memory consumption) | 34.4 |
| **Database** | Focuses on database efficiency (e.g., query latency) | 29.3 |
| **Networking** | Encompasses network behavior, including bandwidth usage or latency | 10.0 |
| **Security** | Relates to authentication, intrusion attempts, or compliance checks | 0.3 |

Table 8: Taxonomy of system domains in the benchmark dataset with their relative proportions. A single time series can belong to multiple domains;

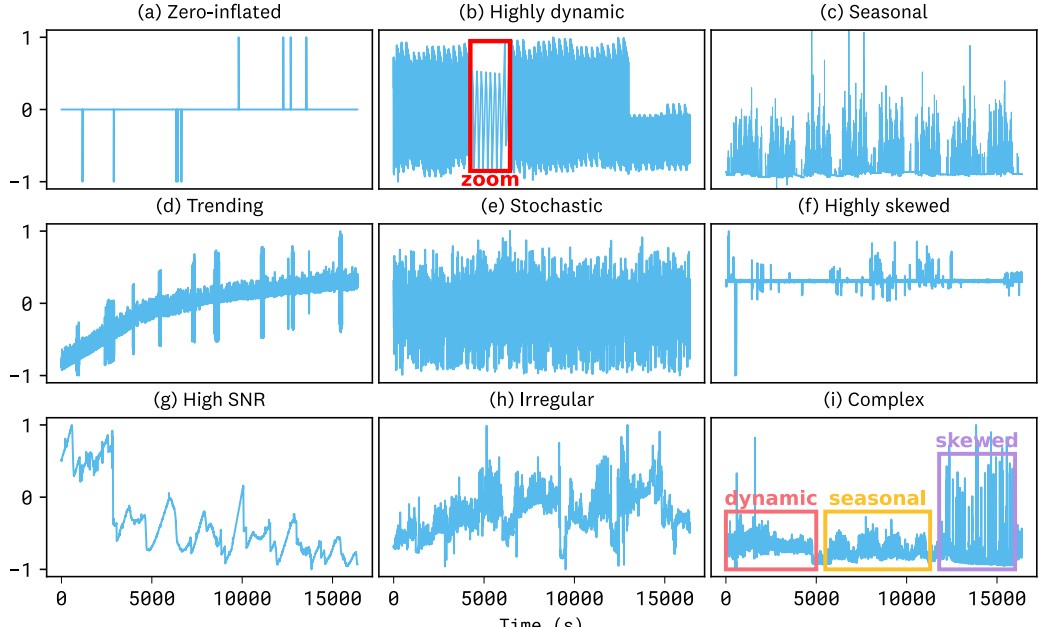

Figure 10: Representative examples from the BOOM, illustrating the unique temporal patterns associated observability data.

padding to ensure quality and comparability. To avoid data leakage in the provided validation split, only one slice is selected per metric, randomly sampled and similar for all the groups. For metrics with more than 100 variates (groups), we randomly subsample 100 to cap input dimensionality. Series are then normalized using the first 90% split of points that will be used for the context windows. We further filter out variates exhibiting abnormal scale change in the test split (final 10% of points) while having constant values in the context window, as these result in degenerate cases that are impossible to forecast meaningfully. This preprocessing results in high-quality, variable-length slices that preserve the structural diversity and challenges of real-world observability time series.

The final benchmark comprises 350 million points across 2,807 metric queries. These series vary widely in sampling frequency, temporal length, and number of variates. Figure 9 illustrates the distribution of series frequencies (left), lengths (middle), and cardinalities (right).

## C.2  Evaluation protocol

### C.2.1  Prediction terms and evaluation windows

Following the GIFT-Eval protocol, we assign a fixed prediction horizon to each time series based on its intrinsic sampling frequency. The specific mapping from frequency to default horizon length is provided in Table 9. This default value defines the *short*-term prediction task.

To define *medium-* and *long*-term prediction tasks, we scale the *short*-term horizon by factors of 10 and 15, respectively. These extended horizons are only applied when they fit entirely within the

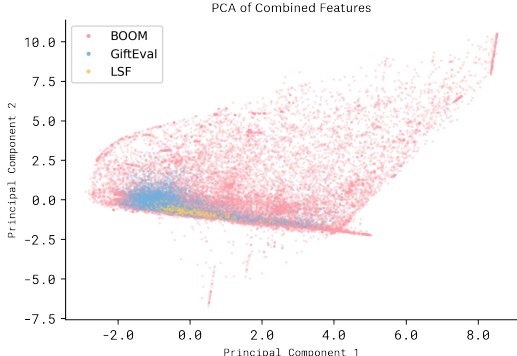

Figure 11: 2D Principal Component Analysis [93] projection of normalized statistical features computed from the Datadog Benchmark and GIFT-Eval datasets. The clear separation between the two distributions highlights a significant shift in underlying time series characteristics.

| Frequency | Monthly (M) | Weekly (W) | Daily (D) | Hourly (H) | Minutely (T) | Secondly (S) |
|---|---|---|---|---|---|---|
| **Prediction Length (steps)** | 12 | 8 | 30 | 48 | 48 | 60 |

Table 9: Mapping from time series frequency to default short-term prediction length.

test window, defined as the final 10% of the time series. This strategy enables evaluation across increasingly challenging forecasting ranges.

For each prediction term, series are evaluated using a rolling, non-overlapping window scheme, where each window has a length equal to the corresponding prediction horizon. This ensures full coverage of the test split while avoiding overlapping forecasts.

The evaluation is hierarchical: if a series qualifies for long-term forecasting (i.e., has enough test points to accommodate the longest horizon), it is also evaluated under the medium- and short-term settings. In total, this procedure yields 7,416 evaluation instances across the 2,807 time series in the benchmark. Each instance corresponds to the average performance over multiple rolling windows within a series for a specific prediction term.

### C.2.2 Evaluation metrics

Following GIFT-Eval, our two main metrics of forecasting accuracy are Mean Absolute Scaled Error (MASE) and Continuous Ranked Probability Score (CRPS).

MASE [77], a point-forecasting score, is defined as

$$\text{MASE} = \frac{\text{MAE}_{\text{model}}}{\text{MAE}_{\text{seasonal naive in-sample}}} \tag{12}$$

with the in-sample Seasonal Naive MAE being defined as

$$\text{MAE}_{\text{seasonal naive in-sample}} = \frac{1}{T-m} \sum_{t=m+1}^{T} Y_t - Y_{t-m} \tag{13}$$

where $T$ is the length of the *training* split of the series, $Y_t$ is the value of the series at time $t$, and $m$ is the seasonal period (typically defined based upon a lookup table according to the time series frequency).

CRPS [78] is scoring rule for probabilistic forecasts, defined with respect to an observation $y$

$$\text{CRPS}(D, y) = \int_{\mathbb{R}} (F_D(x) - H(x - y))^2 dx \tag{14}$$

where $F_D$ is the cumulative distribution function of the forecast distribution $D$ and $H$ is the Heaviside step function. We follow the standard practice of taking the mean weighted quantile loss as a discrete approximation, as described by Park et al. [79].

As in GIFT-Eval, both MASE and CRPS are further normalized by the performance of the Seasonal Naive forecast on the *test* split.

### C.2.3 Inference procedures

To evaluate the comparison models on BOOM, we closely follow the evaluation methodology used in the GIFT-Eval implementation. For models not included in GIFT-Eval, we rely on their official implementations and recommended evaluation procedures. All foundation models are evaluated using a unified context length of 2048. This choice is informed by preliminary experiments showing that a shorter context length (512) leads to a general degradation in performance across models. Therefore, we opt for a relatively large context window (2048) to preserve forecast quality, while ensuring feasibility on available hardware.

To evaluate the zero-shot performance of other foundation models on BOOM, we follow the sampling procedures outlined in their respective manuscripts. For Chronos, we generate 20 samples and use the median prediction as the point forecast. For Moirai, we generate 100 samples, again taking the median, and set the patch size to "auto". For TOTO we generate 256 samples and take the median as the point forecast. TimesFM produces only point forecasts of the mean, which we use directly. In all cases, we compute CRPS with respect to the probabilistic samples and MASE with respect to the point forecast. Since TimesFM and Chronos support only univariate forecasting, we evaluate each variate independently. In contrast, both Moirai and TOTO support joint prediction over groups of related variates.

For the three statistical baselines—AutoARIMA, AutoTheta, and AutoETS—we use the default hyperparameter settings from the statsforecast package, with one exception: for AutoARIMA, we reduce $max_d$ and $max_D$ from 2 to 1 due to frequent numerical instability when $d = D = 2$. Following GIFT-Eval, we set the maximum input length for all statistical models to 1000.

### C.2.4 Aggregation of results

Common practice for aggregating normalized benchmarking statistics is to use the geometric mean [94]. We adopt this approach, with a slight caveat: due to the presence of constant subsequences in observability time series, it's possible for models to achieve zero error on a handful of series in BOOM. As zeros cause geometric means to collapse, we instead use the shifted geometric mean, which is stable in such scenarios [80, 81], for aggregating MASE and CRPS. Specifically, for a metric $m$ across a set of test instances $N$, the shifted geometric mean $\bar{m}_{\text{ShiftedGeom}}$ is defined as

$$\bar{m}_{\text{ShiftedGeom}} = \exp\left(\frac{1}{|N|} \sum_{n \in N} \log(m_n + \varepsilon)\right) + \varepsilon \tag{15}$$

where $\varepsilon = 1\mathrm{e}{-5}$ is a small stabilizing constant.

Evaluating forecasts on observability data using these standard evaluation metrics introduces several numerical instabilities which we must also handle:

**NaN values**: The benchmark includes a small number of zero-inflated series, which can result in invalid CRPS values due to division-by-zero errors. To mitigate this, we apply a generic postprocessing strategy where NaN and infinite values are imputed using the mean CRPS across the remaining series.

**Extremely low naive errors**: Certain flat series exhibit extremely low or even null in-sample MAE, leading to instability when normalizing by the seasonal naive MAE, with potentially exploding normalized values. Rather than discarding these special cases — which are informative for evaluating how models handle anomalous flat behavior — we isolate them into a separate data split. This split is evaluated using simple MAE instead of MASE and non-normalized CRPS. To define this subset, we apply an objective criterion: we include all series where either the MAE for the Seasonal Naive forecast on the test set or the in-sample Seasonal Naive MAE 13 are zero. If any prediction window of any variate within a query satisfies the above conditions, we assign the entire query to the second,

| Metric | Toto | Moirai$_{Small}$ | Moirai$_{Base}$ | Moirai$_{Large}$ | TimesFM$_{2.0}$ | Chronos-Bolt$_{Small}$ | Chronos-Bolt$_{Base}$ | Timer | Time-MoE | VisionTS |
|--------|------|---------|--------|---------|----------|-------------------|------------------|-------|----------|----------|
| MAE ↓ | 0.001 | 0.001 | 0.000 | 0.001 | 0.014 | 0.003 | 0.003 | 0.001 | 0.001 | 0.001 |
| CRPS ↓ | 0.025 | 0.009 | 0.003 | 0.005 | 0.091 | 0.022 | 0.019 | 0.005 | 0.005 | 0.009 |

Table 10: Performance of Toto and other zero-shot models on the BOOM dataset using the subset of near-constant series.

low-variability set, where metrics are computed without normalization. For evaluation, results in this subset are not normalized and thus are aggregated using a simple arithmetic mean.

The outcomes for this subset are reported in Table 10. All the zero-shot models evaluated seem to handle these cases trivially; they all have extremely low MASE. Notably, TOTO's CRPS is slightly elevated relative to the other models. We observe that TOTO seems to produce overly wide prediction intervals in at least some of these flat series, and conjecture that this may be due to the frequent presence of large anomalies in its pretraining data. This is an interesting avenue for further study, as this conservatism may in fact be desirable in downstream anomaly detection use cases where overconfident predictions can lead to false positive detections.

**Taxonomy breakdowns**. For the BOOM, we evaluate model performance across stratified groups defined by the dataset's taxonomy categories. For term (Table 12), metric type (Table 13), and domain (Table 14), we report the shifted geometric mean of the evaluation metric within each group. These results are discussed in detail in Section D.1.

### C.3 BOOMLET

To facilitate research in settings where it is computationally infeasible to evaluate on the full BOOM benchmark, we construct a smaller, representative subset denoted as BOOMLET. This subset is created via uniform sampling over metric queries in BOOM, while additionally prioritizing queries with a relatively large number of variates to ensure sufficient training signal. This selection strategy preserves the distributional properties across key taxonomy dimensions, while also maintaining a practical volume of training data per query. BOOMLET comprises 32 metric queries, encompassing 1,627 variates and approximately 23 million observation points.

## D Results

### D.1 BOOM

In Fig. 12, we present qualitative comparisons across three representative forecasting scenarios to highlight the behavioral differences between TOTO, Chronos, and Moirai. In the first example Ⓐ, features a highly stochastic signal interwoven with complex seasonal components. While Moirai and Chronos models tend to overfit short-term fluctuations—resulting in jagged forecasts and unstable confidence intervals— TOTO effectively identifies and extrapolates the latent seasonal structure, yielding smoother, more coherent trajectories and uncertainty bands that reflect a deeper structural understanding of the series dynamics. Example Ⓑ the target signal exhibits high dynamism with rapidly oscillating structure and sustained amplitude modulations—posing a challenge for long-range temporal modeling. While both Moirai and Chronos models progressively lose phase alignment and dampen their amplitude estimates, TOTO consistently maintains sharp, temporally aligned forecasts with well-calibrated uncertainty, accurately tracking the intricate periodic structure far into the forecast horizon. Finally, example Ⓒ, the target series is characterized by sparse, bursty impulses with high variance across events. Here, although TOTO 's mean prediction does not always precisely capture individual peaks, its predictive distribution faithfully mirrors the underlying spikiness of the series, in stark contrast to Chronos, which collapses to an overconfident flat trajectory.

Table 11 reports the results for all versions and sizes of the zero-shot models.

| Dataset | Metric | Toto | Moirai$_{Small}$ | Moirai$_{Base}$ | Moirai$_{Large}$ | TimesFM$_{2.0}$ | Chronos-Bolt$_{Small}$ | Chronos-Bolt$_{Base}$ | Timer | Time-MoE$_{50M}$ | Time-MoE$_{200M}$ | VisionTS | DLinear | DeepAR | Naive |
|---------|--------|------|---------|--------|---------|----------|-------------------|------------------|-------|-----------|------------|----------|---------|--------|-------|
| BOOM | MASE ↓ | **0.617** | 0.729 | 0.710 | 0.720 | 0.725 | 0.733 | 0.726 | 0.796 | 0.806 | 0.881 | 0.988 | - | - | 1.000 |
| | CRPS ↓ | **0.375** | 0.442 | 0.428 | 0.436 | 0.447 | 0.455 | 0.451 | 0.639 | 0.649 | 0.643 | 0.673 | - | - | 1.000 |
| | Rank ↓ | **2.369** | 4.905 | 4.328 | 4.561 | 5.243 | 5.927 | 5.576 | 9.920 | 9.877 | 9.843 | 10.989 | - | - | 12.631 |
| BOOMLET | MASE ↓ | **0.617** | 0.786 | 0.779 | 0.767 | 0.685 | 0.717 | 0.711 | 0.807 | 0.810 | 0.793 | 0.912 | 0.823 | 0.883 | 1.000 |
| | CRPS ↓ | **0.519** | 0.631 | 0.630 | 0.621 | 0.603 | 0.642 | 0.637 | 0.793 | 0.788 | 0.780 | 0.885 | 0.641 | 0.697 | 1.000 |
| | Rank ↓ | **1.300** | 5.711 | 5.300 | 4.967 | 4.867 | 6.756 | 6.511 | 11.544 | 11.222 | 11.189 | 13.589 | 6.056 | 7.900 | 14.133 |

Table 11: **BOOM results.** Full results across all models evaluated from Table 2. Key: **Best results**, Second-best results.

To better understand the capabilities and limitations of different forecasting models, we conduct a disaggregated evaluation across four major characteristics that describe time series in the BOOM dataset. This analysis enables us to probe how models respond to structural diversity in real-world time series data.

Across all three categorical axes, the TOTO consistently achieves the lowest CRPS, with strong margins over all baselines.

| Real Term | Metric | Toto | Moirai$_{Small}$ | Moirai$_{Base}$ | Moirai$_{Large}$ | TimesFM$_{2.0}$ | Chronos-Bolt$_{Small}$ | Chronos-Bolt$_{Base}$ | Timer | Time-MoE$_{Base}$ | Time-MoE$_{Large}$ | VisionTS | Naive |
|---|---|---|---|---|---|---|---|---|---|---|---|---|---|
| Long | MASE ↓ | **0.688** | 0.795 | 0.780 | 0.799 | 0.817 | 0.813 | 0.798 | 0.809 | 0.886 | 0.950 | 1.026 | 1.000 |
| | CRPS ↓ | **0.424** | 0.482 | 0.473 | 0.491 | 0.522 | 0.528 | 0.519 | 0.661 | 0.724 | 0.694 | 0.698 | 1.000 |
| Medium | MASE ↓ | **0.657** | 0.771 | 0.753 | 0.770 | 0.780 | 0.782 | 0.782 | 0.804 | 0.866 | 0.929 | 1.011 | 1.000 |
| | CRPS ↓ | **0.406** | 0.476 | 0.460 | 0.475 | 0.499 | 0.508 | 0.507 | 0.671 | 0.725 | 0.692 | 0.698 | 1.000 |
| Short | MASE ↓ | **0.535** | 0.670 | 0.627 | 0.626 | 0.619 | 0.638 | 0.632 | 0.779 | 0.704 | 0.794 | 0.947 | 1.000 |
| | CRPS ↓ | **0.318** | 0.399 | 0.370 | 0.369 | 0.359 | 0.368 | 0.365 | 0.597 | 0.541 | 0.570 | 0.640 | 1.000 |

Table 12: Performance comparison of TOTO and other zero-shot models across different **prediction terms**. MASE and CRPS are normalized by the Seasonal Naive forecast and aggregated across tasks using the shifted geometric mean. Key: **Best results**, Second-best results.

| Type | Metric | Toto | Moirai$_{Small}$ | Moirai$_{Base}$ | Moirai$_{Large}$ | TimesFM$_{2.0}$ | Chronos-Bolt$_{Small}$ | Chronos-Bolt$_{Base}$ | Timer | Time-MoE$_{Base}$ | Time-MoE$_{Large}$ | VisionTS | Naive |
|---|---|---|---|---|---|---|---|---|---|---|---|---|---|
| Count | MASE ↓ | 0.687 | 0.814 | 0.795 | 0.813 | 0.919 | 0.883 | 0.880 | 0.663 | **0.652** | 1.035 | 1.220 | 1.000 |
| | CRPS ↓ | **0.317** | 0.370 | 0.353 | 0.372 | 0.403 | 0.403 | 0.402 | 0.662 | 0.651 | 0.698 | 0.603 | 1.000 |
| Distribution | MASE ↓ | **0.658** | 0.741 | 0.724 | 0.729 | 0.745 | 0.759 | 0.753 | 0.890 | 0.878 | 0.877 | 1.034 | 1.000 |
| | CRPS ↓ | **0.382** | 0.434 | 0.422 | 0.428 | 0.440 | 0.452 | 0.446 | 0.608 | 0.604 | 0.596 | 0.674 | 1.000 |
| Gauge | MASE ↓ | **0.583** | 0.720 | 0.686 | 0.700 | 0.706 | 0.706 | 0.696 | 0.721 | 0.760 | 0.890 | 0.922 | 1.000 |
| | CRPS ↓ | **0.382** | 0.471 | 0.444 | 0.456 | 0.466 | 0.469 | 0.463 | 0.658 | 0.694 | 0.703 | 0.672 | 1.000 |
| Rate | MASE ↓ | **0.634** | 0.753 | 0.728 | 0.733 | 0.726 | 0.742 | 0.739 | 0.864 | 0.846 | 0.862 | 1.041 | 1.000 |
| | CRPS ↓ | **0.369** | 0.433 | 0.418 | 0.422 | 0.431 | 0.445 | 0.443 | 0.630 | 0.619 | 0.596 | 0.687 | 1.000 |

Table 13: Performance comparison of TOTO and other zero-shot models across different **metric types**. MASE and CRPS are normalized by the Seasonal Naive forecast and aggregated across tasks using the shifted geometric mean. Key: **Best results**, Second-best results.

| Domain | Metric | Toto | Moirai$_{Small}$ | Moirai$_{Base}$ | Moirai$_{Large}$ | TimesFM$_{2.0}$ | Chronos-Bolt$_{Small}$ | Chronos-Bolt$_{Base}$ | Timer | Time-MoE$_{Base}$ | Time-MoE$_{Large}$ | VisionTS | Naive |
|---|---|---|---|---|---|---|---|---|---|---|---|---|---|
| Application usage | MASE ↓ | **0.639** | 0.747 | 0.721 | 0.730 | 0.736 | 0.748 | 0.748 | 0.871 | 0.863 | 0.884 | 1.042 | 1.000 |
| | CRPS ↓ | **0.378** | 0.440 | 0.422 | 0.430 | 0.441 | 0.452 | 0.451 | 0.636 | 0.633 | 0.611 | 0.691 | 1.000 |
| Database | MASE ↓ | **0.635** | 0.751 | 0.738 | 0.743 | 0.765 | 0.761 | 0.757 | 0.716 | 0.714 | 0.903 | 1.017 | 1.000 |
| | CRPS ↓ | **0.362** | 0.429 | 0.414 | 0.418 | 0.440 | 0.444 | 0.441 | 0.619 | 0.618 | 0.633 | 0.647 | 1.000 |
| Infrastructure | MASE ↓ | **0.568** | 0.692 | 0.650 | 0.670 | 0.679 | 0.678 | 0.663 | 0.728 | 0.791 | 0.847 | 0.863 | 1.000 |
| | CRPS ↓ | **0.391** | 0.476 | 0.446 | 0.462 | 0.471 | 0.474 | 0.466 | 0.655 | 0.713 | 0.710 | 0.666 | 1.000 |
| Networking | MASE ↓ | **0.635** | 0.795 | 0.786 | 0.773 | 0.765 | 0.779 | 0.757 | 0.871 | 0.856 | 0.933 | 1.035 | 1.000 |
| | CRPS ↓ | **0.400** | 0.493 | 0.484 | 0.484 | 0.493 | 0.506 | 0.489 | 0.725 | 0.721 | 0.739 | 0.734 | 1.000 |
| Security | MASE ↓ | **0.682** | 0.741 | 0.739 | 0.736 | 0.717 | 0.734 | 0.729 | 0.828 | 0.770 | 0.776 | 0.924 | 1.000 |
| | CRPS ↓ | **0.476** | 0.505 | 0.504 | 0.504 | 0.525 | 0.539 | 0.535 | 0.664 | 0.625 | 0.629 | 0.735 | 1.000 |

Table 14: Performance comparison of TOTO and other zero-shot models across different **metric domains**. MASE and CRPS are normalized by the Seasonal Naive forecast and aggregated across tasks using the shifted geometric mean. Key: **Best results**, Second-best results.

### D.1.1 BOOMLET

We present results on the BOOMLET subset in Table 11.

### D.2 GIFT-Eval

To provide a comprehensive evaluation of Toto's forecasting capabilities, we benchmarked our model on the GIFT-Eval benchmark [26]. GIFT-Eval is a collection of diverse time series datasets that covers a wide range of domains and characteristics, including:

- Various frequencies (hourly, daily, weekly, monthly, yearly)
- Different domains (energy, traffic, retail, economics, etc.)
- Varying series lengths and forecasting horizons
- Single and multiple seasonality patterns

The benchmark evaluates models using multiple metrics, with particular emphasis on:

- Mean Absolute Scaled Error (MASE): Measures point forecast accuracy relative to a naive forecast
- Continuous Ranked Probability Score (CRPS): Evaluates the quality of probabilistic forecasts
- Overall Rank: Aggregates performance across all datasets

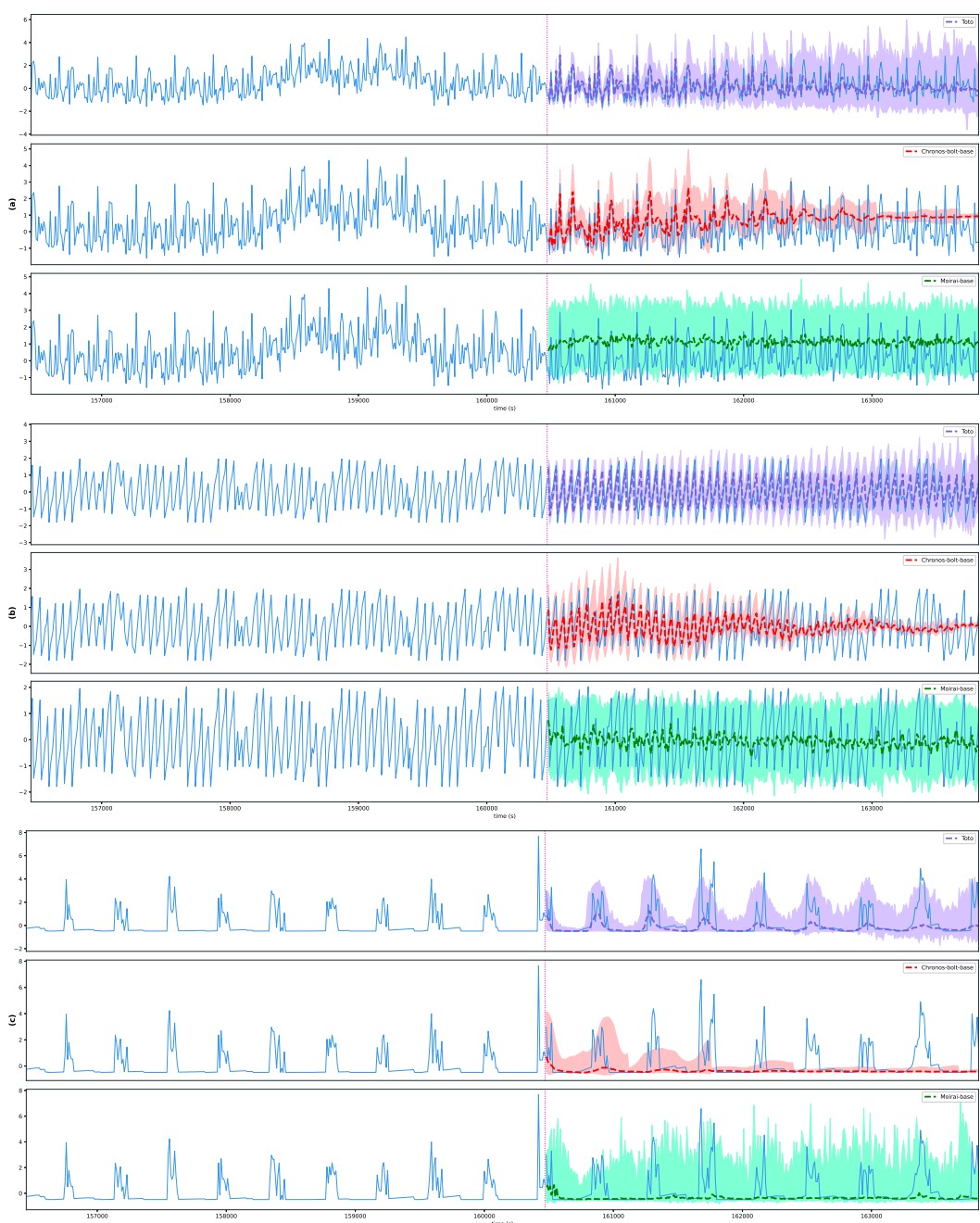

Figure 12: Example of 336-step zero-shot comparative forecasts on the Boom, showing multivariate probabilistic predictions. Solid lines represent ground truth, dashed lines represent median point forecasts, and shaded regions represent 95% prediction intervals.

For models other than TOTO, we report the published numbers from the public leaderboard [3]. For TOTO, we use the same inference settings described in Section C.2.3, with one modification: as GIFT-Eval does impose a maximum context length, we select a length of 4096 (the native context length used in TOTO's pretraining, as described in Table 6).

## D.3 LSF

In addition to our primary evaluations, we also assess the model's performance on the Long Sequence Forecasting (LSF) benchmark datasets—ETTh1, ETTh2, ETTm1, ETTm2, Electricity, and Weather [2]. As noted by

Aksu et al. [1], these datasets are limited in size and diversity, and recent findings [29] suggest that strong supervised baselines can already perform near the upper bound on such benchmarks. This may indicate a saturation point where further gains from foundation models are difficult to observe, rather than a fundamental limitation of the models themselves. Nevertheless, as it remains a widely used legacy benchmark in the literature, we report zero-shot results of TOTO on it to maintain consistency with established practices in the field.

Furthermore we leverage its small scale and constrained use-cases to examine TOTO's capacity to transfer to new datasets and specialized domains by conducting fine-tuning experiments on the training splits of its datasets.

Following standard practice for the LSF benchmark, we report normalized Mean Absolute Error (MAE) and Mean Squared Error (MSE), in order to be able to compare performance across different datasets. We evaluate using forecast lengths of 96, 192, 336, and 720 time steps. Predictions are generated using sliding windows with a stride of 1. For the Electricity dataset, however, we use a stride equal to the prediction length to reduce computational resource requirements. The results are then averaged. We compare TOTO's performance with results reported by recent state-of-the-art time series foundation models, including Moirai [13], VisionTS [24], TimesFM [12], Time-MoE [14], TimeLLM [95], GPT4TS [96], xLSTMTime [97] and other models evaluated in Woo et al. [13] and Das et al. [12]. We display zero-shot and full-shot TOTO results in Table 4 and Table 15 respectively. We also provide additional per prediction length results in Table 16 and Table 17.

Table 4 shows that TOTO consistently delivers the best overall performance across all datasets, achieving the lowest average MAE and MSE, and outperforming other zero-shot baselines on 8 out of 12 evaluation metrics. Its performance is especially strong on ETTm2, Electricity, and Weather, where it continues to excel even in zero-shot scenarios.

|  |  | Zero Shot | | Full Shot | | | | | | | | | | | | | | | |
|---|---|---|---|---|---|---|---|---|---|---|---|---|---|---|---|---|---|---|---|
| Dataset | Metric | Toto | Toto_FT | TimeLLM | GPT4TS | VisionTS_FT | Time-MoE_BaseFT | Time-MoE_LargeFT | Time-MoE_UltraFT | TimesFM* | xLSTMTime | iTransformer | TimesNet | PatchTST | Crossformer | TiDE | DLinear | SCINet | FEDformer |
| ETTh1 | MAE↓ | 0.413 | 0.409 | 0.423 | 0.426 | 0.409 | 0.406 | **0.404** | 0.406 | 0.426 | 0.428 | 0.448 | 0.450 | 0.455 | 0.522 | 0.507 | 0.452 | 0.647 | 0.460 |
|  | MSE↓ | 0.435 | 0.415 | 0.408 | 0.427 | 0.395 | 0.379 | 0.375 | **0.373** | – | 0.408 | 0.454 | 0.458 | 0.469 | 0.529 | 0.541 | 0.456 | 0.747 | 0.440 |
| ETTh2 | MAE↓ | 0.363 | **0.363** | 0.383 | 0.394 | 0.382 | 0.386 | 0.386 | 0.380 | 0.410 | 0.386 | 0.407 | 0.497 | 0.407 | 0.684 | 0.550 | 0.515 | 0.723 | 0.449 |
|  | MSE↓ | 0.340 | 0.339 | **0.334** | 0.354 | 0.336 | 0.346 | 0.361 | **0.334** | – | 0.346 | 0.383 | 0.414 | 0.387 | 0.942 | 0.611 | 0.559 | 0.954 | 0.437 |
| ETTm1 | MAE↓ | 0.378 | **0.357** | 0.372 | 0.383 | 0.367 | 0.381 | 0.371 | 0.373 | 0.388 | 0.371 | 0.410 | 0.406 | 0.400 | 0.495 | 0.419 | 0.407 | 0.481 | 0.452 |
|  | MSE↓ | 0.396 | 0.349 | 0.329 | 0.352 | 0.338 | 0.345 | **0.322** | 0.329 | – | 0.347 | 0.407 | 0.400 | 0.387 | 0.513 | 0.419 | 0.403 | 0.486 | 0.448 |
| ETTm2 | MAE↓ | 0.303 | **0.291** | 0.313 | 0.326 | 0.319 | 0.335 | 0.332 | 0.334 | 0.334 | 0.310 | 0.332 | 0.333 | 0.326 | 0.611 | 0.404 | 0.401 | 0.537 | 0.349 |
|  | MSE↓ | 0.267 | **0.244** | 0.251 | 0.266 | 0.261 | 0.271 | 0.284 | 0.277 | – | 0.254 | 0.288 | 0.291 | 0.281 | 0.757 | 0.358 | 0.350 | 0.571 | 0.305 |
| Electricity | MAE↓ | 0.243 | **0.234** | 0.252 | 0.263 | 0.249 | – | – | – | – | 0.250 | 0.270 | 0.295 | 0.304 | 0.334 | 0.344 | 0.300 | 0.365 | 0.327 |
|  | MSE↓ | 0.161 | **0.152** | 0.158 | 0.167 | 0.156 | – | – | – | – | 0.157 | 0.178 | 0.193 | 0.216 | 0.244 | 0.252 | 0.212 | 0.268 | 0.214 |
| Weather | MAE↓ | 0.245 | **0.233** | 0.257 | 0.270 | 0.262 | 0.275 | 0.273 | 0.280 | – | 0.255 | 0.278 | 0.287 | 0.281 | 0.315 | 0.320 | 0.317 | 0.363 | 0.360 |
|  | MSE↓ | 0.224 | **0.206** | 0.225 | 0.237 | 0.227 | 0.236 | 0.234 | 0.250 | – | 0.222 | 0.258 | 0.259 | 0.259 | 0.259 | 0.271 | 0.265 | 0.292 | 0.309 |
| Mean | MAE↓ | 0.324 | **0.314** | 0.333 | 0.344 | 0.331 | – | – | – | – | 0.333 | 0.358 | 0.378 | 0.362 | 0.494 | 0.424 | 0.399 | 0.519 | 0.400 |
|  | MSE↓ | 0.304 | **0.284** | 0.284 | 0.300 | 0.286 | – | – | – | – | 0.289 | 0.328 | 0.336 | 0.333 | 0.541 | 0.409 | 0.374 | 0.553 | 0.359 |
| Best Count |  |  | 8 | 1 | 0 | 0 | 0 | 2 | 2 | 0 | 0 | 0 | 0 | 0 | 0 | 0 | 0 | 0 | 0 |

Table 15: Full-Shot comparison of models on the LSF benchmark, with TOTO's Zero-Shot result in the first data column. *TimesFM only reports values for MAE on ETTh1, ETTh2, ETTm1, and ETTm2 after fine-tuning.
Key: **Best results**, Second-best results. "Best Count" row reports the number of times each model attains the best result for a given dataset-metric pair.

Furthermore, Table 15 shows that even when starting from a strong SOTA baseline, TOTO's performance improves with fine-tuning, showing it can achieve full-shot SOTA results and adapt to new domains with limited data. This highlights TOTO's robustness and versatility as a foundation model for a wide range of time-series forecasting tasks.

**Full-shot results on LSF benchmarks** We conduct fine-tuning experiments on Toto following similar procedure delineated by [98] and [13]. The full-shot results for each dataset, comparing fine-tuned and zero-shot performance, are reported in Table 15.

**Results** Our experimental results demonstrate that when finetuned, denoted as TOTO$_{FT}$, achieves state-of-the-art performance on 3 out of 6 datasets in the LSF benchmark—specifically, ETTm2, Electricity, and Weather—where it outperforms all other models on both MAE and MSE metrics. Additionally, TOTO$_{FT}$ achieves the best MAE score on ETTm1 and ETTh2, although it does not lead on MSE for those datasets. Compared to its zero-shot counterpart, TOTO$_{FT}$ consistently improves both MAE and MSE metrics across most datasets, with particularly notable gains in ETTm1 (MAE: 0.378 → 0.357, MSE: 0.396 → 0.349) and ETTm2 (MAE: 0.303 → 0.291, MSE: 0.267 → 0.244). Overall, TOTO$_{FT}$ ranks first in 8 out of 12 metric-dataset pairs, outperforming all other models, including both zero-shot and full-shot baselines. Notably, it also delivers the best overall performance on the benchmark, achieving the lowest average MAE (0.314) and MSE (0.284). These results underscore the effectiveness of fine-tuning in enhancing Toto's predictive performance, establishing TOTO$_{FT}$ as the new SOTA model on the LSF benchmark. In addition, this demonstrates that Toto is a robust foundation model, adaptable to a wide range of downstream datasets, including those from entirely new domains, making it a versatile choice for time-series forecasting tasks.

A closer examination of the results reveals that while Toto$_{FT}$ achieves state-of-the-art performance on most datasets, the effectiveness of fine-tuning varies across them. Fine-tuning proves especially beneficial on ETTm1, ETTm2, and Weather, where it significantly enhances model predictions. In contrast, the improvements on ETTh1 are more modest, and for ETTh2, fine-tuning yields no notable gains—potentially due to the relatively small size of these datasets. Moreover, even though fine-tuning generally improves performance over the original TOTO model, TOTO$_{FT}$ does not outperform other full-shot models on ETTh1.

Additional details on zero-shot and full-shot results per prediction length are displayed in Table 17

| Dataset | Prediction Length | Metric | Toto | Moirai$_{Small}$ | Moirai$_{Base}$ | Moirai$_{Large}$ | VisionTS | TIME-MoE$_{Base}$ | TIME-MoE$_{Large}$ | TIME-MoE$_{Ultra}$ |
|---------|-------------------|--------|------|------------------|-----------------|------------------|----------|-------------------|--------------------|--------------------|
| | | | | | | *Zero Shot* | | | | |
| ETTh1 | 96 | MAE ↓ | 0.381 | 0.402 | 0.402 | 0.398 | 0.383 | 0.381 | 0.382 | **0.379** |
| | | MSE ↓ | 0.382 | 0.375 | 0.384 | 0.380 | 0.353 | 0.357 | 0.350 | **0.349** |
| | 192 | MAE ↓ | 0.408 | 0.419 | 0.429 | 0.434 | 0.410 | **0.404** | 0.412 | 0.413 |
| | | MSE ↓ | 0.428 | 0.399 | 0.425 | 0.440 | 0.392 | **0.384** | 0.388 | 0.395 |
| | 336 | MAE ↓ | **0.422** | 0.429 | 0.450 | 0.474 | 0.423 | 0.434 | 0.430 | 0.453 |
| | | MSE ↓ | 0.457 | 0.412 | 0.456 | 0.514 | **0.407** | 0.411 | 0.411 | 0.447 |
| | 720 | MAE ↓ | **0.440** | 0.444 | 0.473 | 0.568 | 0.441 | 0.477 | 0.455 | 0.462 |
| | | MSE ↓ | 0.472 | 0.413 | 0.470 | 0.705 | **0.406** | 0.449 | 0.427 | 0.457 |
| ETTh2 | 96 | MAE ↓ | **0.310** | 0.334 | 0.327 | 0.325 | 0.328 | 0.359 | 0.354 | 0.352 |
| | | MSE ↓ | 0.273 | 0.281 | 0.277 | 0.287 | **0.271** | 0.305 | 0.302 | 0.292 |
| | 192 | MAE ↓ | **0.356** | 0.373 | 0.374 | 0.367 | 0.367 | 0.386 | 0.385 | 0.379 |
| | | MSE ↓ | 0.339 | 0.340 | 0.340 | 0.347 | **0.328** | 0.351 | 0.364 | 0.347 |
| | 336 | MAE ↓ | 0.387 | 0.393 | 0.401 | 0.393 | **0.381** | 0.418 | 0.425 | 0.419 |
| | | MSE ↓ | 0.374 | 0.362 | 0.371 | 0.377 | **0.345** | 0.391 | 0.417 | 0.406 |
| | 720 | MAE ↓ | **0.400** | 0.416 | 0.426 | 0.421 | 0.422 | 0.454 | 0.496 | 0.447 |
| | | MSE ↓ | **0.375** | 0.380 | 0.394 | 0.404 | 0.388 | 0.419 | 0.537 | 0.439 |
| ETTm1 | 96 | MAE ↓ | **0.333** | 0.383 | 0.360 | 0.363 | 0.347 | 0.368 | 0.357 | 0.341 |
| | | MSE ↓ | 0.320 | 0.404 | 0.335 | 0.353 | 0.341 | 0.338 | 0.309 | **0.281** |
| | 192 | MAE ↓ | 0.364 | 0.402 | 0.379 | 0.380 | 0.360 | 0.388 | 0.381 | **0.358** |
| | | MSE ↓ | 0.371 | 0.435 | 0.366 | 0.376 | 0.360 | 0.353 | 0.346 | **0.305** |
| | 336 | MAE ↓ | 0.388 | 0.416 | 0.394 | 0.395 | **0.374** | 0.413 | 0.408 | 0.395 |
| | | MSE ↓ | 0.408 | 0.462 | 0.391 | 0.399 | 0.377 | 0.381 | 0.373 | **0.369** |
| | 720 | MAE ↓ | 0.426 | 0.437 | 0.419 | 0.417 | **0.405** | 0.493 | 0.477 | 0.472 |
| | | MSE ↓ | 0.485 | 0.490 | 0.434 | 0.432 | **0.416** | 0.504 | 0.475 | 0.469 |
| ETTm2 | 96 | MAE ↓ | **0.237** | 0.282 | 0.269 | 0.260 | 0.282 | 0.291 | 0.286 | 0.288 |
| | | MSE ↓ | **0.172** | 0.205 | 0.195 | 0.189 | 0.228 | 0.201 | 0.197 | 0.198 |
| | 192 | MAE ↓ | **0.280** | 0.318 | 0.303 | 0.300 | 0.305 | 0.334 | 0.322 | 0.312 |
| | | MSE ↓ | 0.232 | 0.261 | 0.247 | 0.247 | 0.262 | 0.258 | 0.250 | 0.235 |
| | 336 | MAE ↓ | **0.320** | 0.355 | 0.333 | 0.334 | 0.328 | 0.373 | 0.375 | 0.348 |
| | | MSE ↓ | 0.290 | 0.319 | 0.291 | 0.295 | 0.293 | 0.324 | 0.337 | 0.293 |
| | 720 | MAE ↓ | 0.375 | 0.410 | 0.377 | 0.386 | **0.370** | 0.464 | 0.461 | 0.428 |
| | | MSE ↓ | 0.372 | 0.415 | 0.355 | 0.372 | **0.343** | 0.488 | 0.480 | 0.427 |
| Electricity | 96 | MAE ↓ | **0.213** | 0.299 | 0.248 | 0.242 | 0.266 | - | - | - |
| | | MSE ↓ | **0.129** | 0.205 | 0.158 | 0.152 | 0.177 | - | - | - |
| | 192 | MAE ↓ | **0.229** | 0.310 | 0.263 | 0.259 | 0.277 | - | - | - |
| | | MSE ↓ | **0.145** | 0.220 | 0.174 | 0.171 | 0.188 | - | - | - |
| | 336 | MAE ↓ | **0.247** | 0.323 | 0.278 | 0.278 | 0.296 | - | - | - |
| | | MSE ↓ | **0.163** | 0.236 | 0.191 | 0.192 | 0.207 | - | - | - |
| | 720 | MAE ↓ | **0.282** | 0.347 | 0.307 | 0.313 | 0.337 | - | - | - |
| | | MSE ↓ | **0.206** | 0.270 | 0.229 | 0.236 | 0.256 | - | - | - |
| Weather | 96 | MAE ↓ | **0.179** | 0.212 | 0.203 | 0.208 | 0.257 | 0.214 | 0.213 | 0.211 |
| | | MSE ↓ | **0.149** | 0.173 | 0.167 | 0.177 | 0.220 | 0.160 | 0.159 | 0.157 |
| | 192 | MAE ↓ | **0.223** | 0.250 | 0.241 | 0.249 | 0.275 | 0.260 | 0.266 | 0.256 |
| | | MSE ↓ | **0.192** | 0.216 | 0.209 | 0.219 | 0.244 | 0.210 | 0.215 | 0.208 |
| | 336 | MAE ↓ | **0.265** | 0.282 | 0.276 | 0.292 | 0.299 | 0.309 | 0.322 | 0.290 |
| | | MSE ↓ | **0.245** | 0.260 | 0.256 | 0.277 | 0.280 | 0.274 | 0.291 | 0.255 |
| | 720 | MAE ↓ | **0.312** | 0.322 | 0.323 | 0.350 | 0.337 | 0.405 | 0.400 | 0.397 |
| | | MSE ↓ | **0.310** | 0.320 | 0.321 | 0.365 | 0.330 | 0.418 | 0.415 | 0.405 |
| **Best Count** | | | **29** | **0** | **0** | **0** | **11** | **2** | **0** | **6** |

Table 16: Zero-Shot Comparison of different models with TOTO on the LSF benchmark datasets for each prediction length. Non-TOTO values are reproduced from published tables.

Key: **Best results**, Second-best results. "Best Count" row reports the number of times each model attains the best result for a given metric.

| Dataset | Prediction Length | Metric | Zero Shot | | Full Shot | | | | | | | | | | | | | | | |
|---|---|---|---|---|---|---|---|---|---|---|---|---|---|---|---|---|---|---|---|---|
| | | | Toto | Toto$_{FT}$ | TimeLLM | GPT4TS | VisionTS$_{FT}$ | Time-MoE$_{Base}$FT | Time-MoE$_{Large}$FT | Time-MoE$_{Ultra}$FT | TimesFM* | xLSTMTime | iTransformer | TimesNet | PatchTST | Crossformer | TiDE | DLinear | SCINet | FEDformer |
| ETTh1 | 96 | MAE↓ | 0.381 | 0.374 | 0.392 | 0.397 | 0.376 | 0.373 | 0.371 | **0.365** | 0.398 | 0.395 | 0.405 | 0.402 | 0.419 | 0.448 | 0.464 | 0.400 | 0.599 | 0.419 |
| | | MSE↓ | 0.382 | 0.364 | 0.362 | 0.376 | 0.347 | 0.345 | 0.335 | **0.323** | - | 0.368 | 0.386 | 0.384 | 0.414 | 0.423 | 0.479 | 0.386 | 0.654 | 0.376 |
| | 192 | MAE↓ | 0.408 | 0.402 | 0.418 | 0.418 | 0.400 | 0.396 | 0.400 | **0.391** | 0.424 | 0.416 | 0.441 | 0.429 | 0.445 | 0.474 | 0.492 | 0.432 | 0.631 | 0.448 |
| | | MSE↓ | 0.428 | 0.409 | 0.427 | 0.416 | 0.385 | 0.374 | 0.374 | **0.359** | - | 0.401 | 0.458 | 0.436 | 0.460 | 0.471 | 0.525 | 0.437 | 0.719 | 0.420 |
| | 336 | MAE↓ | 0.422 | 0.418 | **0.398** | 0.433 | 0.415 | 0.412 | 0.412 | 0.418 | 0.436 | 0.437 | 0.458 | 0.469 | 0.466 | 0.546 | 0.515 | 0.459 | 0.659 | 0.465 |
| | | MSE↓ | 0.457 | 0.436 | 0.430 | 0.442 | 0.407 | 0.389 | 0.390 | 0.388 | - | 0.422 | 0.487 | 0.491 | 0.501 | 0.570 | 0.565 | 0.481 | 0.778 | 0.459 |
| | 720 | MAE↓ | 0.440 | 0.440 | 0.457 | 0.456 | 0.443 | 0.443 | **0.433** | 0.450 | 0.445 | 0.465 | 0.491 | 0.500 | 0.488 | 0.621 | 0.558 | 0.516 | 0.699 | 0.507 |
| | | MSE↓ | 0.472 | 0.454 | 0.442 | 0.477 | 0.439 | 0.410 | **0.402** | 0.425 | - | 0.441 | 0.503 | 0.521 | 0.500 | 0.653 | 0.594 | 0.519 | 0.836 | 0.506 |
| ETTh2 | 96 | MAE↓ | 0.310 | **0.309** | 0.328 | 0.342 | 0.328 | 0.340 | 0.335 | 0.338 | 0.356 | 0.333 | 0.349 | 0.374 | 0.348 | 0.584 | 0.440 | 0.387 | 0.621 | 0.397 |
| | | MSE↓ | 0.273 | 0.272 | **0.268** | 0.285 | 0.269 | 0.276 | 0.278 | 0.274 | - | 0.273 | 0.297 | 0.340 | 0.302 | 0.745 | 0.400 | 0.333 | 0.707 | 0.358 |
| | 192 | MAE↓ | 0.356 | **0.355** | 0.375 | 0.389 | 0.374 | 0.371 | 0.373 | 0.370 | 0.400 | 0.378 | 0.400 | 0.414 | 0.400 | 0.656 | 0.509 | 0.476 | 0.689 | 0.439 |
| | | MSE↓ | 0.339 | **0.338** | 0.329 | 0.354 | 0.332 | 0.331 | 0.345 | 0.330 | - | 0.340 | 0.380 | 0.402 | 0.388 | 0.877 | 0.528 | 0.477 | 0.860 | 0.429 |
| | 336 | MAE↓ | 0.387 | **0.386** | 0.409 | 0.407 | 0.395 | 0.402 | 0.402 | 0.396 | 0.428 | 0.403 | 0.428 | 0.452 | 0.433 | 0.731 | 0.571 | 0.541 | 0.744 | 0.487 |
| | | MSE↓ | 0.374 | **0.372** | 0.368 | 0.373 | 0.356 | 0.373 | 0.384 | 0.362 | - | 0.373 | 0.426 | 0.452 | 0.426 | 1.043 | 0.643 | 0.594 | 1.000 | 0.496 |
| | 720 | MAE↓ | 0.400 | **0.400** | 0.420 | 0.441 | 0.430 | 0.431 | 0.437 | 0.417 | 0.457 | 0.430 | 0.459 | 0.462 | 0.446 | 0.763 | 0.679 | 0.657 | 0.838 | 0.474 |
| | | MSE↓ | 0.375 | 0.374 | 0.372 | 0.406 | 0.390 | 0.404 | 0.409 | **0.370** | - | 0.398 | 0.491 | 0.478 | 0.431 | 1.104 | 0.874 | 0.831 | 1.249 | 0.463 |
| ETTm1 | 96 | MAE↓ | 0.333 | 0.313 | 0.334 | 0.346 | 0.322 | 0.334 | 0.325 | **0.323** | 0.345 | 0.335 | 0.368 | 0.375 | 0.367 | 0.426 | 0.387 | 0.372 | 0.438 | 0.419 |
| | | MSE↓ | 0.320 | 0.278 | 0.272 | 0.292 | 0.281 | 0.286 | 0.264 | **0.256** | - | 0.286 | 0.334 | 0.338 | 0.329 | 0.404 | 0.364 | 0.345 | 0.418 | 0.379 |
| | 192 | MAE↓ | 0.364 | 0.345 | 0.358 | 0.372 | 0.353 | 0.358 | 0.350 | **0.343** | 0.374 | 0.361 | 0.391 | 0.387 | 0.385 | 0.451 | 0.404 | 0.389 | 0.450 | 0.441 |
| | | MSE↓ | 0.371 | 0.328 | 0.310 | 0.332 | 0.322 | 0.307 | 0.295 | **0.281** | - | 0.329 | 0.377 | 0.374 | 0.367 | 0.450 | 0.398 | 0.380 | 0.439 | 0.426 |
| | 336 | MAE↓ | 0.388 | 0.368 | 0.384 | 0.394 | 0.379 | 0.390 | 0.376 | **0.374** | 0.397 | 0.379 | 0.420 | 0.411 | 0.410 | 0.515 | 0.425 | 0.413 | 0.485 | 0.459 |
| | | MSE↓ | 0.408 | 0.364 | 0.352 | 0.366 | 0.356 | 0.354 | **0.323** | 0.326 | - | 0.358 | 0.426 | 0.410 | 0.399 | 0.532 | 0.428 | 0.413 | 0.490 | 0.445 |
| | 720 | MAE↓ | 0.426 | 0.403 | 0.411 | 0.421 | 0.413 | 0.445 | 0.435 | 0.452 | 0.436 | 0.411 | 0.459 | 0.450 | 0.439 | 0.589 | 0.461 | 0.453 | 0.550 | 0.490 |
| | | MSE↓ | 0.485 | 0.426 | **0.383** | 0.417 | 0.391 | 0.433 | 0.409 | 0.454 | - | 0.416 | 0.491 | 0.478 | 0.454 | 0.666 | 0.487 | 0.474 | 0.595 | 0.543 |
| ETTm2 | 96 | MAE↓ | 0.237 | **0.227** | 0.253 | 0.262 | 0.256 | 0.265 | 0.259 | 0.273 | 0.263 | 0.250 | 0.264 | 0.267 | 0.259 | 0.366 | 0.305 | 0.292 | 0.377 | 0.287 |
| | | MSE↓ | 0.172 | **0.158** | 0.161 | 0.173 | 0.169 | 0.172 | 0.169 | 0.183 | - | 0.164 | 0.180 | 0.187 | 0.175 | 0.287 | 0.207 | 0.193 | 0.286 | 0.203 |
| | 192 | MAE↓ | 0.280 | **0.269** | 0.293 | 0.301 | 0.294 | 0.306 | 0.295 | 0.301 | 0.309 | 0.288 | 0.309 | 0.309 | 0.302 | 0.492 | 0.364 | 0.362 | 0.445 | 0.328 |
| | | MSE↓ | 0.232 | **0.212** | 0.219 | 0.229 | 0.225 | 0.228 | 0.223 | 0.223 | - | 0.218 | 0.250 | 0.249 | 0.241 | 0.414 | 0.290 | 0.284 | 0.399 | 0.269 |
| | 336 | MAE↓ | 0.320 | **0.306** | 0.329 | 0.341 | 0.334 | 0.345 | 0.341 | 0.339 | 0.349 | 0.322 | 0.348 | 0.351 | 0.343 | 0.542 | 0.422 | 0.427 | 0.591 | 0.366 |
| | | MSE↓ | 0.290 | **0.263** | 0.271 | 0.286 | 0.278 | 0.281 | 0.293 | 0.278 | - | 0.271 | 0.311 | 0.321 | 0.305 | 0.597 | 0.377 | 0.369 | 0.637 | 0.325 |
| | 720 | MAE↓ | 0.375 | **0.362** | 0.379 | 0.401 | 0.392 | 0.424 | 0.433 | 0.424 | 0.415 | 0.380 | 0.407 | 0.403 | 0.400 | 1.042 | 0.524 | 0.522 | 0.735 | 0.415 |
| | | MSE↓ | 0.372 | **0.344** | 0.352 | 0.378 | 0.372 | 0.403 | 0.451 | 0.425 | - | 0.361 | 0.412 | 0.408 | 0.402 | 1.730 | 0.558 | 0.554 | 0.960 | 0.421 |
| Electricity | 96 | MAE↓ | 0.213 | **0.207** | 0.224 | 0.238 | 0.218 | - | - | - | - | 0.221 | 0.240 | 0.272 | 0.285 | 0.314 | 0.329 | 0.282 | 0.345 | 0.308 |
| | | MSE↓ | 0.129 | **0.123** | 0.131 | 0.139 | 0.126 | - | - | - | - | 0.128 | 0.148 | 0.168 | 0.195 | 0.219 | 0.237 | 0.197 | 0.247 | 0.193 |
| | 192 | MAE↓ | 0.229 | **0.224** | 0.241 | 0.251 | 0.237 | - | - | - | - | 0.243 | 0.253 | 0.289 | 0.289 | 0.322 | 0.330 | 0.285 | 0.355 | 0.315 |
| | | MSE↓ | 0.145 | **0.142** | 0.152 | 0.153 | 0.144 | - | - | - | - | 0.150 | 0.162 | 0.184 | 0.199 | 0.231 | 0.236 | 0.196 | 0.257 | 0.201 |
| | 336 | MAE↓ | 0.247 | **0.239** | 0.248 | 0.266 | 0.256 | - | - | - | - | 0.259 | 0.269 | 0.300 | 0.305 | 0.337 | 0.344 | 0.301 | 0.369 | 0.329 |
| | | MSE↓ | 0.163 | **0.155** | 0.160 | 0.169 | 0.162 | - | - | - | - | 0.166 | 0.178 | 0.198 | 0.215 | 0.246 | 0.249 | 0.209 | 0.269 | 0.214 |
| | 720 | MAE↓ | 0.282 | **0.266** | 0.298 | 0.297 | 0.286 | - | - | - | - | 0.276 | 0.317 | 0.320 | 0.337 | 0.363 | 0.373 | 0.333 | 0.390 | 0.355 |
| | | MSE↓ | 0.206 | 0.187 | 0.192 | 0.206 | 0.192 | - | - | - | - | **0.185** | 0.225 | 0.220 | 0.256 | 0.280 | 0.284 | 0.245 | 0.299 | 0.246 |
| Weather | 96 | MAE↓ | 0.179 | **0.165** | 0.201 | 0.212 | 0.192 | - | - | - | - | 0.187 | 0.214 | 0.220 | 0.218 | 0.230 | 0.261 | 0.255 | 0.306 | 0.296 |
| | | MSE↓ | 0.149 | **0.134** | 0.147 | 0.162 | 0.142 | - | - | - | - | 0.144 | 0.174 | 0.172 | 0.177 | 0.158 | 0.202 | 0.196 | 0.221 | 0.217 |
| | 192 | MAE↓ | 0.223 | **0.211** | 0.234 | 0.248 | 0.238 | - | - | - | - | 0.236 | 0.254 | 0.261 | 0.259 | 0.277 | 0.298 | 0.296 | 0.340 | 0.336 |
| | | MSE↓ | 0.192 | **0.177** | 0.189 | 0.204 | 0.191 | - | - | - | - | 0.192 | 0.221 | 0.219 | 0.225 | 0.206 | 0.242 | 0.237 | 0.261 | 0.276 |
| | 336 | MAE↓ | 0.265 | **0.253** | 0.279 | 0.286 | 0.282 | - | - | - | - | 0.272 | 0.296 | 0.306 | 0.297 | 0.335 | 0.335 | 0.335 | 0.378 | 0.380 |
| | | MSE↓ | 0.245 | **0.225** | 0.262 | 0.254 | 0.246 | - | - | - | - | 0.237 | 0.278 | 0.280 | 0.278 | 0.272 | 0.287 | 0.283 | 0.309 | 0.339 |
| | 720 | MAE↓ | 0.312 | **0.302** | 0.316 | 0.337 | 0.337 | - | - | - | - | 0.326 | 0.349 | 0.359 | 0.348 | 0.418 | 0.386 | 0.381 | 0.427 | 0.428 |
| | | MSE↓ | 0.310 | **0.288** | 0.304 | 0.326 | 0.328 | - | - | - | - | 0.313 | 0.358 | 0.365 | 0.354 | 0.398 | 0.351 | 0.345 | 0.377 | 0.403 |
| Best Count | | | 0 | 30 | 3 | 0 | 0 | 0 | 4 | 9 | 0 | 1 | 0 | 0 | 0 | 0 | 0 | 0 | 0 | 0 |

Table 17: Full-Shot Comparison of different models with TOTO on the LSF benchmark datasets for each prediction length, with TOTO's Zero-Shot result in the first data column. Key: **Best results**, Second-best results. "Best Count" row reports the number of times each model attains the best result for a given metric.

## D.4 Computational Efficiency

We have conducted an empirical study to evaluate the computational efficiency of recent time series foundation models, which we present in Table 18. All experiments were performed on a single NVIDIA A100 (40 GB) GPU using synthetic multivariate time series. Each model was profiled under identical settings, with a context length of 2,048 and a prediction horizon of 480 for variable number of variates. We include comparisons against both multivariate (Moirai-Base) and univariate models (Time-MoE-50M, Chronos-Bolt-Base, and TimesFM-500M).

Table 18: Model Benchmark Results

| # Variates | Model | Wall Time (ms) | CUDA Time (ms) | Peak Memory (MB) | Total FLOPs (GFLOPs) |
|---|---|---|---|---|---|
| 10 | TOTO | $652.8 \pm 12.6$ | $37.0 \pm 0.0$ | $721.0 \pm 0.0$ | $\mathbf{121.6 \pm 0.0}$ |
| | TOTO (no KV cache) | $585.0 \pm 7.9$ | $45.8 \pm 4.1$ | $733.6 \pm 0.1$ | $885.2 \pm 0.0$ |
| | Moirai | $\mathbf{148.1 \pm 2.4}$ | $\mathbf{19.1 \pm 1.7}$ | $\mathbf{489.8 \pm 0.0}$ | $167.6 \pm 0.0$ |
| | Time-MoE | $1569.0 \pm 150.1$ | $361.2 \pm 4.1$ | $5341.2 \pm 0.0$ | $3449.9 \pm 0.0$ |
| | Chronos | $831.6 \pm 25.2$ | $93.7 \pm 0.0$ | $925.5 \pm 0.0$ | $2162.5 \pm 0.0$ |
| | TimesFM | $579.5 \pm 44.9$ | $109.6 \pm 0.1$ | $2023.6 \pm 0.0$ | $5188.1 \pm 0.0$ |
| 30 | TOTO | $627.2 \pm 10.4$ | $43.8 \pm 0.4$ | $934.9 \pm 0.0$ | $\mathbf{364.7 \pm 0.0}$ |
| | TOTO (no KV cache) | $628.2 \pm 55.5$ | $75.6 \pm 5.5$ | $971.9 \pm 0.3$ | $2655.5 \pm 0.0$ |
| | Moirai | $\mathbf{284.1 \pm 204.7}$ | $\mathbf{86.4 \pm 4.3}$ | $1221.6 \pm 0.0$ | $642.1 \pm 0.0$ |
| | Time-MoE | $1924.7 \pm 22.6$ | $970.4 \pm 5.5$ | $15036.3 \pm 0.0$ | $10349.8 \pm 0.0$ |
| | Chronos | $925.8 \pm 22.1$ | $179.7 \pm 8.4$ | $1135.4 \pm 0.0$ | $6487.5 \pm 0.0$ |
| | TimesFM | $698.6 \pm 18.1$ | $280.3 \pm 8.6$ | $2200.9 \pm 0.0$ | $15564.3 \pm 0.0$ |
| 50 | TOTO | $687.5 \pm 72.6$ | $\mathbf{50.7 \pm 2.9}$ | $1148.3 \pm 0.0$ | $\mathbf{607.8 \pm 0.0}$ |
| | TOTO (no KV cache) | $627.1 \pm 55.4$ | $99.9 \pm 7.7$ | $1209.3 \pm 0.0$ | $4425.8 \pm 0.0$ |
| | Moirai | $\mathbf{303.1 \pm 23.6}$ | $186.4 \pm 1.5$ | $2642.0 \pm 0.0$ | $1302.2 \pm 0.0$ |
| | Time-MoE | $3239.9 \pm 252.8$ | $1572.0 \pm 0.7$ | $24730.3 \pm 0.0$ | $17249.6 \pm 0.0$ |
| | Chronos | $1029.6 \pm 14.8$ | $266.5 \pm 18.2$ | $1335.4 \pm 0.5$ | $10812.5 \pm 0.0$ |
| | TimesFM | $662.0 \pm 9.2$ | $441.4 \pm 11.6$ | $\mathbf{2383.2 \pm 0.0}$ | $25940.5 \pm 0.0$ |
| 70 | TOTO | $\mathbf{667.1 \pm 12.4}$ | $57.6 \pm 4.5$ | $1361.6 \pm 0.0$ | $\mathbf{850.9 \pm 0.0}$ |
| | TOTO (no KV cache) | $669.8 \pm 55.9$ | $126.5 \pm 2.6$ | $1447.4 \pm 0.0$ | $6196.1 \pm 0.0$ |
| | Moirai | $454.8 \pm 16.1$ | $346.0 \pm 2.9$ | $4763.1 \pm 0.0$ | $2148.0 \pm 0.0$ |
| | Time-MoE | $8612.0 \pm 248.8$ | $3464.1 \pm 168.3$ | $30394.6 \pm 423.5$ | $37882.8 \pm 1968.5$ |
| | Chronos | $1193.4 \pm 44.0$ | $333.4 \pm 10.7$ | $\mathbf{1543.3 \pm 0.0}$ | $15137.6 \pm 0.0$ |
| | TimesFM | $842.0 \pm 15.2$ | $606.6 \pm 7.7$ | $2555.2 \pm 0.0$ | $36316.6 \pm 0.0$ |
| 90 | TOTO | $\mathbf{712.1 \pm 48.6}$ | $\mathbf{66.9 \pm 0.0}$ | $1575.9 \pm 0.0$ | $\mathbf{1094.0 \pm 0.0}$ |
| | TOTO (no KV cache) | $714.8 \pm 91.3$ | $156.8 \pm 4.5$ | $1685.5 \pm 0.0$ | $7966.5 \pm 0.0$ |
| | Moirai | $667.2 \pm 5.5$ | $562.1 \pm 1.0$ | $7577.0 \pm 0.0$ | $3179.4 \pm 0.0$ |
| | Time-MoE | $8985.5 \pm 729.0$ | $3315.5 \pm 157.5$ | $36581.7 \pm 2532.6$ | $36010.0 \pm 1572.7$ |
| | Chronos | $1209.9 \pm 69.8$ | $427.9 \pm 7.8$ | $\mathbf{1750.2 \pm 0.0}$ | $19462.6 \pm 0.0$ |
| | TimesFM | $1124.7 \pm 36.1$ | $786.3 \pm 8.5$ | $2727.2 \pm 0.0$ | $46692.8 \pm 0.0$ |
| 150 | TOTO | $\mathbf{708.3 \pm 8.8}$ | $\mathbf{89.8 \pm 0.8}$ | $\mathbf{2223.2 \pm 0.0}$ | $\mathbf{1823.3 \pm 0.0}$ |
| | TOTO (no KV cache) | $714.6 \pm 31.5$ | $257.5 \pm 5.2$ | $2406.4 \pm 0.0$ | $13277.4 \pm 0.0$ |
| | Moirai | $1830.3 \pm 122.1$ | $1488.5 \pm 0.7$ | $20193.6 \pm 0.0$ | $7387.2 \pm 0.0$ |
| | Time-MoE | $11903.6 \pm 677.6$ | $5109.4 \pm 271.8$ | $33639.9 \pm 2419.7$ | $56227.9 \pm 3128.8$ |
| | Chronos | $1435.2 \pm 25.1$ | $670.4 \pm 4.3$ | $2375.2 \pm 0.0$ | $32437.6 \pm 0.0$ |
| | TimesFM | $1699.8 \pm 7.7$ | $1239.3 \pm 5.1$ | $3256.4 \pm 0.0$ | $77821.4 \pm 0.0$ |
| 200 | TOTO | $\mathbf{710.2 \pm 16.1}$ | $\mathbf{100.9 \pm 4.7}$ | $\mathbf{2757.5 \pm 0.0}$ | $\mathbf{2431.1 \pm 0.0}$ |
| | TOTO (no KV cache) | $747.7 \pm 71.8$ | $340.7 \pm 6.1$ | $2998.2 \pm 0.0$ | $17703.2 \pm 0.0$ |
| | Moirai | $3097.1 \pm 380.8$ | $2316.1 \pm 1.3$ | $35487.6 \pm 0.0$ | $12170.0 \pm 0.0$ |
| | Time-MoE | $15668.0 \pm 152.2$ | $7828.7 \pm 3.0$ | $30352.4 \pm 0.0$ | $86490.8 \pm 0.0$ |
| | Chronos | $1799.4 \pm 23.9$ | $890.3 \pm 7.0$ | $2897.4 \pm 0.0$ | $43250.2 \pm 0.0$ |
| | TimesFM | $2284.4 \pm 24.1$ | $1665.3 \pm 3.7$ | $3698.9 \pm 0.0$ | $103761.8 \pm 0.0$ |
| 300 | TOTO | $\mathbf{775.2 \pm 15.4}$ | $\mathbf{132.8 \pm 3.6}$ | $\mathbf{3821.7 \pm 0.0}$ | $\mathbf{3646.7 \pm 0.0}$ |
| | TOTO (no KV cache) | $805.1 \pm 25.3$ | $505.8 \pm 5.3$ | $4187.8 \pm 0.0$ | $26554.9 \pm 0.0$ |
| | Moirai | OOM | OOM | OOM | OOM |
| | Time-MoE | $16590.2 \pm 254.8$ | $9698.9 \pm 0.2$ | $31517.0 \pm 0.0$ | $108459.7 \pm 0.0$ |
| | Chronos | $2583.3 \pm 23.6$ | $1290.2 \pm 4.5$ | $3935.1 \pm 0.0$ | $64875.2 \pm 0.0$ |
| | TimesFM | $3302.4 \pm 22.0$ | $2445.4 \pm 5.9$ | $4569.9 \pm 0.0$ | $155642.8 \pm 0.0$ |

To isolate the fundamental computational cost of each architecture, we evaluate simple forward passes generating a single output sample. This configuration reflects the core operation dominating training and fine-tuning, where repeated forward evaluations occur at scale, and thus provides a standardized measurement of training efficiency. Many models, including TOTO, increase the number of samples at inference time to boost predictive accuracy; this analysis does not address such test-time scaling.

TOTO demonstrates consistently strong computational efficiency and scalability compared to other models, with the lowest FLOPs across all variate counts. It achieves the lowest or second-lowest wall times across all variate counts, maintaining stable performance even as the input dimensionality increases from 10 to 300 variates. Starting from 150 variates onward, TOTO outperforms Moirai on all metrics, including wall time and CUDA time; the gap grows significantly with more variates. Even against univariate competitors, where cross-variate

interaction is ignored, TOTO provides better performance, demonstrating its highly optimized architecture and efficient memory use. We note that TOTO uses KV caching, while other models do not appear to implement this. For additional transparency, we also share TOTO statistics with KV caching turned off and note that the general trend remains the same.

# E    Ablations

We evaluate the contribution of various architectural components of the TOTO model by systematically disabling one component at a time and measuring the relative performance degradation. The full Toto model serves as the control, and each variant's performance is presented relative to this baseline in Table 19. All models in the ablation study, including the control, were trained for 75,000 steps (a subset of the full-length training of the TOTO base model).

| Model | Best NLL Loss (% increase) ↓ |
|---|---|
| **Control** | **0.0%** |
| No Variate-wise Attention | 1.6% |
| No Robust Loss | 11.1% |
| No Student-T Mixture | 27.2% |
| No Causal Scaling | 27.3% |

Table 19: Relative change in NLL on held-out observability pretraining data when removing key design features of the TOTO architecture.

To compare performance between the different arms of the experiment, we look at NLL loss on a held-out validation split of the observability portion of the pretraining data. This summarizes the output distribution and gives us a single performance metric to compare both point forecasting and probabilistic forecasting. For each model, we pick the checkpoint with lowest NLL throughout the training run (evaluating on the validation set every 5,000 steps).

The results reveal that removing key modeling elements significantly impacts performance. Disabling Causal Scaling leads to the largest degradation, with an increase of 27.3% in NLL when we replace the causal scaler with a naive global scaler. Replacing the Student-T mixture model with a single Student-T output causes a similar NLL increase of 27.2%. Interestingly, removing the robust loss component and optimizing NLL alone actually leads to a *worse* overall NLL, with an 11.1% increase; we speculate this is because the robust loss stabilizes the training, as discussed in Section 3.1. Finally, removing the variate-wise attention (i.e. making all the attention layers time-wise while holding the parameter count constant) leads to a more modest increase in NLL of 1.6%.

# F    Impact statement

In developing TOTO, we followed a structured approach to ensure responsible development, focusing on identifying, assessing, and mitigating potential risks associated with the use of our model. Given that TOTO specifically generates time series forecasts, the potential harms are considerably lower compared to language, image, or other more general-purpose models. Our primary focus was ensuring the accuracy and reliability of the forecasts generated by TOTO, which are crucial for maintaining and optimizing infrastructure and application performance.

The BOOM benchmark provides numerical time series generated from observability metrics, which we view as a valuable resource to the broader time series research community. Each series has an associate high-level application label and no other metadata and contains no PII.

