# OpenReview forum: "This Time is Different: An Observability Perspective on Time Series Foundation Models"
_NeurIPS.cc/2025/Conference — NeurIPS 2025 poster_

### Official Review · Reviewer_RNqe · 2025-06-27

**Clarity:** 2
**Significance:** 3
**Originality:** 3
**Rating:** 5
**Confidence:** 4

**Summary:**

The paper proposes a time series forecasting foundation model (TOTO) to account for the challenges of observability time series data, including Patch-based causal instance normalization, Student-T mixture model head. At the same time, the paper proposes a large-scale benchmark (Boom) consisting of 350 million observations. Extensive experiments show that ToTo achieves good prediction results on Boom and GIFT-Eval benchmarks.

**Questions:**

1. The UNiTS model also utilizes Attention for variable correlation modeling. Please explain the differences compared to your approach. MoIRAI also employs a Student-T Mixture for probability modeling, but the Student-T Mixture can only fit specific distributions. Please clarify how this differs from TOTO.
2. TOTO incorporates variable modeling during pre-training. Since different datasets have different variables:
    a) How do you ensure that the number of variables in each batch of time-series data remains the same?
    b) Could varying numbers of variables across batches lead to difficulties in model pre-training?
3. In Table 2, I believe it is necessary to evaluate smaller models to verify TOTO’s predictive performance. Simply comparing it with zero-shot Foundation Models is unfair because other Foundation Models were pre-trained on very limited observational time-series data, whereas TOTO used a much larger amount.
4. Table 3 lacks comparisons with other Foundation Models such as VisionTS and Time-MOE.
5. There is a lack of analytical experiments, ablation studies, and hyperparameter sensitivity analysis for the model.
6. The main experiments on Foundation Models are all zero-shot evaluations. It would be beneficial to fine-tune TOTO and observe the effects under different levels of fine-tuning.

**Ethical Concerns:**

["NO or VERY MINOR ethics concerns only"]

**Final Justification:**

The author provided clear responses to all my queries, covering TOTO's advantages over other base models and fine-tuning improvements. Overall, I recommend increasing the score.

**Limitations:**

Yes

**Quality:**

3

**Strengths And Weaknesses:**

Strengths:
1. The paper proposes a time series forecasting foundation model for real-world observation data.
2. The paper proposes a large-scale time series benchmark consisting of a large amount of real-world observation data.
3. ToTo achieves good prediction results on Boom and GIFT-Eval benchmarks.

Weaknesses:
1. The primary module proposed in the method section has been used in other Foundation Models, such as variable association modeling and Student-T mixture model Head.
2. The experiment is imperfect, lacking model analysis experiments, ablation experiments, and parameter sensitivity analysis.

---

> ### Author Rebuttal · Authors · 2025-07-30
>
> Thank you for your review and thoughtful questions.
>
> > The primary module proposed in the method section has been used in other Foundation Models, such as variable association modeling and Student-T mixture model Head.
>
> > MoIRAI also employs a Student-T Mixture for probability modeling, but the Student-T Mixture can only fit specific distributions. Please clarify how this differs from TOTO.
>
> >The UNiTS model also utilizes Attention for variable correlation modeling. Please explain the differences compared to your approach.
>
> While we agree there are similarities between Toto’s architecture and other recent works, let us attempt to clarify which contributions we feel are meaningful and novel:
>
> **Student-T Mixture Model (SMM)**
> While other authors have used mixture models (e.g. Moirai), to our knowledge **Toto is the first time series foundation model to use a Student-T Mixture Model head**.
>
> We cite the prior art from Moirai, and took it into consideration when designing our mixture head; however the two approaches differ in crucial and meaningful ways. Moirai does *not* use a Student-T mixture model. As stated in their paper, Moirai uses a fixed mixture of four different distributions [1]. The Moirai authors suggest that these four individual components were chosen heuristically to match specific common cases within time series datasets.
>
> By contrast, Toto's mixture consists of ***only*** Student-T distributions, where the number of components, $k$, is a hyperparameter that we optimize ($k$=24 in our selected model). This is fundamentally a more general and less heuristic approach. We derive it from first principles: mixtures of certain distributions (including Gaussians and Student-T’s) are universal approximators for ***any probability density function*** when using a sufficiently large number of mixture components [2, 3]. We choose the SMM (rather than GMM for example) based on its superior performance with heavy-tailed distributions [4], which frequently appear in the observability domain.
>
> We also adopt a novel loss formulation which leads to significantly better convergence when training with our mixture model head. Alongside NLL, we jointly optimize the SMM with Robust Loss [4] to concentrate the mean towards strong point predictions.
>
> Based on this feedback, we will revise the discussion in our Related Works section to make these distinctions clearer.
>
> **Causal patch-based instance normalization**
> This is a novel form of normalization which dramatically improves performance in our ablation study (Figure 3B). It does this by preserving the causality of our autoregressive decoder-style model, which is critical for highly nonstationary time series (where future information leakage can cause severe overfitting).
>
> **Proportional factorized attention**
> In Section 3.1, we cite previous demonstrations of factorized attention [5-7]. Our novel contribution introduces an asymmetrical ratio of layers, with the ratio of timewise to variate-wise attention operations as a tunable hyperparameter. While this is a relatively simple idea, it represents a meaningful improvement in computational efficiency. Our design is motivated by the observation that channel-independent models such as TimesFM [8] and Chronos [9] still achieve competitive performance on multivariate benchmarks and, therefore, we expect diminishing returns on FLOPs spent on variate-wise attention operations.
>
> We will add a citation and discussion of UNiTS to make this distinction clearer.
>
> >The experiment is imperfect, lacking model analysis experiments, ablation experiments, and parameter sensitivity analysis.
>
> We do perform an ablation experiment to test the contribution of major architectural choices. The results are reported in Figure 3, and we provide methodological details in Appendix E in the supplemental materials.
>
> We describe our hyperparameter selection procedure, which included an extensive sweep (see Table 5), in Appendix A.6 in the supplemental materials. Unfortunately, a full sensitivity analysis is prohibitively costly given the hyperparameter search space. We can, however, add to the paper a parallel coordinate plot or other data visualization for the configurations evaluated during our hyperparameter sweep.
>
> >TOTO incorporates variable modeling during pre-training. Since different datasets have different variables: a) How do you ensure that the number of variables in each batch of time-series data remains the same? b) Could varying numbers of variables across batches lead to difficulties in model pre-training?
>
> Thanks for the question; it highlights an implementation detail which we mention briefly in Appendix A.2.1 but currently do not discuss in the main body. In pretraining, we use a fixed batch dimension of 32 variates and use document masking to pack multiple unrelated multivariate series in a single batch item. For series with more than 32 variates, we split these into multiple batch items.
>
> At inference time, because the variate-wise attention layers are permutation invariant, we are able to scale up to as many variates as will fit into memory without needing any explicit extrapolation techniques. For example, in GIFT-Eval there are series with up to 321 variates, while BOOM contains series with up to 100 variates.
>
> > In Table 2, I believe it is necessary to evaluate smaller models to verify TOTO’s predictive performance. Simply comparing it with zero-shot Foundation Models is unfair because other Foundation Models were pre-trained on very limited observational time-series data, whereas TOTO used a much larger amount.
>
> We do compare against deep-learning-based full-shot methods on GIFT-Eval, which is a multi-domain benchmark that is not primarily focused on observability data. We find that Toto’s zero-shot forecasts outperform full shot and small baseline models even when these competing models have been explicitly trained on the training splits of the datasets in GIFT-Eval (see Table 3).
>
> To provide further empirical support, we have added evaluations on a subset of BOOM (what we call “BOOMLET” in the paper) using DLinear and DeepAR. These are two smaller full-shot models commonly used as baselines in the field.  Interestingly, DLinear does outperform some other foundation models, including Chronos and Time-MoE. However, Toto still achieves the best overall performance by a wide margin on BOOMLET (we will update Table 2 accordingly):
>
> |model|MASE|CRPS|Rank|
> |:----|:----|:----|:----|
> |*Toto*|*0.617*|*0.519*|*1.300*|
> |TimesFM|0.685|0.603|4.844|
> |Moirai|0.767|0.621|4.911|
> |*DLinear*|*0.823*|*0.641*|*6.044*|
> |Chronos|0.711|0.637|6.411|
> |*DeepAR*|*0.883*|*0.697*|*7.822*|
>
> We followed the same training setup proposed by the GIFT-Eval authors [10] for full-shot evaluation with hyperparameter tuning.
>
> > Table 3 lacks comparisons with other Foundation Models such as VisionTS and Time-MOE.
>
> Table 3 is a reproduction of the GIFT-Eval leaderboard which includes our results. In the interest of saving space, we only printed the top several models in each category (zero-shot, full-shot, and baseline). With respect to VisionTS and Time-MoE:
>
> - VisionTS is not competitive with the top models on GIFT-Eval, with MASE of 0.775, CRPS of 0.638, and Rank of 20.464 as of the time of our submission.
>
> - Time-MoE has not submitted results to the GIFT-Eval leaderboard, and thus was not included in Table 3 (which as mentioned is a reproduction of the public leaderboard; we did not rerun models other than Toto).
>
> We are happy to add a supplemental table with the complete GIFT-Eval leaderboard if the reviewer believes it to be necessary.
>
> >  The main experiments on Foundation Models are all zero-shot evaluations. It would be beneficial to fine-tune TOTO and observe the effects under different levels of fine-tuning.
>
> We did perform a fine-tuning experiment using the training portions of the LSF datasets, which we discuss in Section 5. The procedure and results are outlined in Appendix D.3, and summarized in Table 16. Our results show that finetuning results in an improvement Toto’s zero-shot performance, and achieves state-of-the-art performance over other full-shot methods. For convenience, we reproduce an excerpt of Table 16 showing the Toto's finetuned performance on the LSF datasets against the next-best full-shot model (TimeLLM):
>
> |Dataset|Metric|Toto (Zero-shot)|Toto (Fintuned)|TimeLLM (full-shot)|
> |:---:|:---:|:----|:----|:----|
> |ETTh1|MAE|0.413|**0.409**|0.423|
> ||MSE|0.435|0.415|**0.408**|
> |ETTh2|MAE|**0.363**|**0.363**|0.383|
> ||MSE|0.340|**0.339**|0.334|
> |ETTm1|MAE|0.378|**0.357**|0.372|
> ||MSE|0.396|0.349|**0.329**|
> |ETTm2|MAE|0.303|**0.291**|0.313|
> ||MSE|0.267|**0.244**|0.251|
> |Electricity|MAE|0.242|**0.233**|0.252|
> ||MSE|0.158|**0.150**|0.158|
> |Weather|MAE|0.245|**0.233**|0.257|
> ||MSE|0.224|**0.206**|0.225|
> |Avg.|MAE|0.324|**0.314**|0.333|
> ||MSE|0.303|**0.284**|0.284|
>
> ### **Please do let us know if there are any remaining concerns that prevent you from increasing your score.**
>
> [1] Unified Training of Universal Time Series Forecasting Transformers. ICML  2024
> [2] Mixture models and em. In Pattern Recognition and Machine Learning, pages 423–495. Springer, New York, 1 edition, 2006
> [3] Adaptive Mixture of Student-T Distributions As a Flexible Candidate Distribution for Efficient Simulation. Journal of Statistical Software, Jan. 2009
> [4] Robust mixture modelling using the t distribution. Statistics and Computing, 10(4):339–348, 2000.
> [5] A general and adaptive robust loss function. CVPR 2019
> [6] Crossformer: Transformer utilizing cross-dimension dependency for multivariate time series forecasting. ICLR 2023
> [7] Msa transformer. ICML 2021
> [8] Vivit: A video vision transformer. ICCV 2021
> [9] A decoder-only foundation model for time-series forecasting. ICML 2024.
> [10] Chronos: Learning the language of time series. TMLR 2024
> [11] Gift-eval: A benchmark for general time series forecasting model evaluation. arXiv:2410.10393.

---

> > ### Comment · Reviewer_RNqe · 2025-08-05
> >
> > Thank you for the detailed rebuttal. Most of my questions are resolved, and I have just one minor question remaining: during pre-training, the number of input variables is fixed at 32.
> >
> > 1. Why do you choose 32 variables? Is there a specific reason for this choice, or do experiments show that 32 performs best?
> > 2. For datasets that originally have fewer than 32 variables, do you apply a duplication strategy to meet the input requirement?

---

> > > ### Author Response · Authors · 2025-08-05
> > >
> > > Thank you for taking the time to review our rebuttal. In regards to your follow-up questions:
> > >
> > > > Why do you choose 32 variables? Is there a specific reason for this choice, or do experiments show that 32 performs best?
> > >
> > > We identified 32 as a reasonably large variate dimension that still allows us to achieve a desired minimum batch size on the available hardware (our training data is of shape `Batch x Variates x Time Steps`, and memory grows linearly with the batch dimension but quadratically with the variate dimension; See Appendix A.2.1 Complexity Analysis). A study of the effect of different variate dimensions would be a natural direction for further work on Toto.
> > >
> > > > For datasets that originally have fewer than 32 variables, do you apply a duplication strategy to meet the input requirement?
> > >
> > > In cases where we have datasets with fewer than 32 variates, we pack together multiple unrelated time series into the same training item. We assign each group of related variates a unique ID, and use a block-diagonal attention mask [1] in the variate-wise attention layers to prevent variates from unrelated time series from attending to each other.
> > >
> > > This is similar to the technique used by Moirai. It is also very similar to the “sequence packing” that is commonly used in LLM pretraining [2, 3], except that we pack along the variate dimension rather than the sequence dimension.
> > >
> > > We mention this packing approach in passing in Appendix A.2.1; however, we are happy to add a more detailed appendix addressing both of these points comprehensively.
> > >
> > >
> > > 1. Krell, Mario Michael, et al. Efficient Sequence Packing without Cross-Contamination: Accelerating Large Language Models without Impacting Performance. arXiv:2107.02027, 5 Oct. 2022.
> > > 2. Raffel, Colin, et al. “Exploring the Limits of Transfer Learning with a Unified Text-to-Text Transformer.” Journal of Machine Learning Research, vol. 21, no. 140, 2020, pp. 1–67.
> > > 3. Grattafiori, Aaron, et al. The Llama 3 Herd of Models. arXiv:2407.21783, arXiv, 23 Nov. 2024.

---

> > > > ### Author Response · Authors · 2025-08-07
> > > >
> > > > Please do let us know if our reply has addressed your outstanding questions; we're happy to clarify further if needed.

---

> > > > > ### Comment · Reviewer_RNqe · 2025-08-08
> > > > >
> > > > > Thanks to the author for the reply. I have no further questions and I will increase the scores.

---

> > > > > > ### Author Response · Authors · 2025-08-09
> > > > > >
> > > > > > We're glad we were able to address your concerns. Thank you again for taking the time to review our work and for engaging in the discussion period.

---

### Official Review · Reviewer_WAXQ · 2025-06-29

**Clarity:** 2
**Significance:** 3
**Originality:** 2
**Rating:** 4
**Confidence:** 4

**Summary:**

This paper proposes a pre-trained model for time series forecasting in the observability domain. It further introduces BOOM, a large-scale benchmark of 350 million observations derived from observability time series, exceeding the scale of the current largest zero-shot benchmark, GIFT-Eval.

**Questions:**

1. What is the lookback window size used for the LSF benchmark?

**Ethical Concerns:**

["NO or VERY MINOR ethics concerns only"]

**Final Justification:**

The concerns regarding certain aspects of novelty, ablation study, and efficiency have been addressed.

**Limitations:**

Yes

**Quality:**

3

**Strengths And Weaknesses:**

Strengths:
1. This work is valuable to the community by focusing on foundation models for observability time series and further contributes by open-sourcing the constructed dataset to support reproducibility and future research.
2. Extensive experiments conducted on the LSF, BOOM, and GIFT-Eval benchmarks demonstrate the strong generalization ability of TOTO across short-, medium-, and long-term forecasting horizons.

Weaknesses:
1. The technical novelty of this work appears to be limited. First, several proposed components—such as Proportional Factorized Attention and the Student-T Mixture Model—have already been employed in prior foundation models. For instance, introducing a fixed ratio to balance variate-wise and time-wise MHA within a Transformer decoder represents only incremental innovation. Similarly, the Student-T Mixture Model closely resembles the Student’s t-distribution used in Moirai [1], and the method does not present clear distinctions from existing probabilistic forecasting approaches. Second, these components seem broadly applicable to general time series data, and it remains very unclear how they are specifically tailored to the unique characteristics of observability time series.
2. The ablation experiments and analyses of individual components are overly simplistic. Merely removing them is insufficient to fully demonstrate the critical role each plays in the overall architecture.
3. A comprehensive efficiency comparison with existing foundation models such as Moirai and Time-MoE is lacking.

[1] Woo G, Liu C, Kumar A, et al. Unified training of universal time series forecasting transformers. ICML, 2024.

---

> ### Author Rebuttal · Authors · 2025-07-30
>
> We thank the reviewer for their comments and provide clarifying discussion as well as new empirical support.
>
> ## Novelty & Specialization to Observability
> First, there were concerns raised around the novelty of Toto’s main components as well as their specific applicability to observability time series. We respectfully contend that our use of the **Student-T Mixture Model (SMM)** and the **Hybrid NLL/Robust Loss** for time series foundation models is novel; furthermore, the **Causal Patch-based Instance Normalization** algorithm is entirely novel. And while factorized attention has prior art, our specific **Proportional Factorized Attention** design offers meaningful computational efficiency gains tailored to high-cardinality observability data. We’ve organized our discussion per component to clarify what we feel makes them both novel and important for observability.
>
> ### Student-T Mixture Model (SMM)
>
> **Novelty and comparison to Moirai:** While other authors have used mixture models (e.g. Moirai), **to our knowledge Toto is the first time series foundation model to use a Student-T Mixture Model head.** This is a nontrivial improvement and generalization of the mixture model used in Moirai. As stated in their paper, Moirai uses a fixed mixture of four different distributions [1]. The Moirai authors suggest that these four individual components were chosen heuristically to match specific common cases within time series datasets.
>
> By contrast, our model utilizes a mixture consisting of ***only*** Student-T distributions, where the number of components, $k$, is a hyperparameter that we optimize ($k=24$ in our selected model). This is fundamentally a more general and less heuristic approach. We derive our approach from first principles: mixtures of certain distributions (including Gaussians and Student-T’s) are universal approximators for ***any probability density function*** when using a sufficiently large number of mixture components [2, 3]. We choose the SMM (rather than GMM for example) based on its superior performance with heavy-tailed distributions [4].
>
> In contrast to Moirai and other prior work, we also adopt a **novel loss formulation** which leads to significantly better convergence when training with our mixture model head. Alongside NLL, we jointly optimize the SMM with Robust Loss [4] to concentrate the mean towards accurate point predictions.
>
> **Significance for observability:** As we show in Section 4.3, observability time series have challenging and diverse characteristics. The SMM’s expressive power is more important the further away we get from simple, well-behaved distributions. We use SMM rather than a GMM since we find that its heavy tails make it significantly more robust to outliers during training. Finally, the Robust Loss was chosen (rather than L1 or L2 loss) precisely because the presence of outliers in the training data makes these less-robust losses unstable.
>
> ### Causal patch-based instance normalization
>
> **Novelty:**  To improve generalization across varying input scales, instance normalization is commonly applied prior to embedding time series data. However, computing normalization statistics from the entire series would leak information from future time steps; it violates the model’s autoregressive causality and results in poor performance from overfitting. To the best of our knowledge, this is a completely novel approach, and it yields highly non-trivial improvements, as demonstrated in our ablation.
>
> **Significance for observability:**  Preserving causality is especially critical in data which is highly nonstationary, like observability data (due to frequent regime changes and anomalies). The more the distribution shifts over time, the more future information is leaked by global statistics, which hurts generalization performance. Our ablation (Figure 3B) on observability data shows that causal patch-based normalization is the single most impactful architectural improvement of Toto.
>
> ### Proportional factorized attention
>
> **Novelty**: You correctly point out that factorized attention is not novel, and we do not make such a claim. While we agree that introducing an asymmetrical ratio of layers is a relatively simple idea, it nevertheless represents a meaningful improvement in computational efficiency of our model. Our asymmetric design is motivated by the observation that channel-independent models such as TimesFM [5] and Chronos [6] still achieve competitive performance on multivariate benchmarks and, therefore, we expect diminishing returns on FLOPs spent on variate-wise attention operations.
>
> **Significance for observability:** Optimizing the variate-mixing of our model is particularly important for observability data, which typically has higher cardinality than other datasets. For example, our BOOM benchmark has a median number of variates of 60 (vs. 1 for GIFT-Eval and LSF).
>
> We will revise the discussion in our Related Works and Model Architecture sections  to make all of these points clearer.
>
> ### Contribution of BOOM
>
> Beyond the technical contributions described above, we would like to re-emphasize the significant contribution of BOOM, the first large-scale, comprehensive forecasting benchmark within the observability domain. In terms of absolute scale, it contains nearly twice as many points as GIFT-Eval; and as we show in Section 4.3, its distributional characteristics are significantly different from those of GIFT-Eval and LSF.
>
> ## Other feedback / new results
>
> > The ablation experiments and analyses of individual components are overly simplistic. Merely removing them is insufficient to fully demonstrate the critical role each plays in the overall architecture.
>
> We chose a “leave-one-out” ablation design as this allows us to isolate each novel architecture component, while saving significant compute resources vs. a full grid search. This methodology is commonly used in recent foundation model literature; for example, similar leave-one-out ablations are used by Moirai [1], Time-MoE [7], UniTS [8], PowerPM [9], and others.
>
> Furthermore, we describe our hyperparameter selection procedure, which included an extensive sweep, in Appendix A.6 in the supplemental materials.
>
> > A comprehensive efficiency comparison with existing foundation models such as Moirai and Time-MoE is lacking.
>
> Thank you for this feedback, we have conducted an empirical study of inference cost to help address this. We profile several models on synthetic data using a single A100-40GB, with a context length of 2,048 and a prediction length of 480, across 10, 50, 150, and 300 variates. We include comparisons against both multivariate (**Moirai-Base**) and univariate (**Time-MoE-50M**, **Chronos-Bolt-Base**, **TimesFM-500M**) models.
>
> Toto consistently has the lowest FLOPs across all variate counts. Starting from 150 variates onward, Toto outperforms Moirai on all metrics; the gap grows significantly with more variates. Even against univariate competitors, where cross-variate interaction is ignored, Toto provides better performance, demonstrating its highly optimized architecture and efficient memory use. We note that Toto uses KV caching, while other models do not appear to implement this. For additional transparency, we also share Toto statistics with KV caching turned off and note that the general trend remains the same.
>
> The results below show mean and std. dev. across 10 replications for each number of variates. (**best,** *second-best*)
>
> |# Variates| Model | Wall Time (ms) | CUDA Time (ms) | Peak Memory (MB) | Total GFLOPs |
> |-----|:---:|:---:|:---:|:---:|:---:|
> |**10**|**Toto**|652.8±12.6|*37.0±0.0*|*721.0±0.0*|**121.6±0.0**|
> ||**Toto (no kvcache)**|585.0±7.9|45.8±4.1|733.6±0.1|885.2±0.0|
> ||**Moirai**|**148.1±2.4**|**19.1±1.7**|**489.8±0.0**|*167.6±0.0*|
> ||**TimeMoE**|1569.0±150.1|361.2±4.1|5341.2±0.0|3449.9±0.0|
> ||**Chronos**|831.6±25.2|93.7±0.0|925.5±0.0|2162.5±0.0|
> ||**TimesFM**|*579.5±44.9*|109.6±0.1|2023.6±0.0|5188.1±0.0|
> |**150**|**Toto**|**708.3±8.8**|**89.8±0.8**|**2223.2±0.0**|**1823.3±0.0**|
> ||**Toto (no kvcache)**|*714.6±31.5*|*257.5±5.2*|2406.4±0.0|13277.4±0.0|
> ||**Moirai**|1830.3±122.1|1488.5±0.7|20193.6±0.0|*7387.2±0.0*|
> ||**TimeMoE**|11903.6±677.6|5109.4±271.8|33639.9±2419.7|56227.9±3128.8|
> ||**Chronos**|1435.2±25.1|670.4±4.3|*2375.2±0.0*|32437.6±0.0|
> ||**TimesFM**|1699.8±7.7|1239.3±5.1|3256.4±0.0|77821.4±0.0|
> |**300**|**Toto**|**775.2±15.4**|**132.8±3.6**|**3821.7±0.0**|**3646.7±0.0**|
> ||**Toto (no kvcache)**|*805.1±25.3*|*505.8±5.3*|4187.8±0.0|*26554.9±0.0*|
> ||**Moirai**|OOM|OOM|OOM|OOM|
> ||**TimeMoE**|16590.2±254.8|9698.9±0.2|31517.0±0.0|108459.7±0.0|
> ||**Chronos**|2583.3±23.6|1290.2±4.5|*3935.1±0.0*|64875.2±0.0|
> ||**TimesFM**|3302.4±22.0|2445.4±5.9|4569.9±0.0|155642.8±0.0|
>
> >What is the lookback window size used for the LSF benchmark?
>
> We used a fixed lookback window size of 2,048. This is similar to Moirai, which uses lookback windows of 2,000 or 4,000 depending on the dataset.
>
> ### **Please do let us know if there are any remaining concerns that prevent you from increasing your score.**
>
> [1] Unified Training of Universal Time Series Forecasting Transformers. ICML 2024
> [2] Mixture models and em. In Pattern Recognition and Machine Learning, pages 423–495. Springer, New York, 1 edition, 2006
> [3] Adaptive Mixture of Student-T Distributions As a Flexible Candidate Distribution for Efficient Simulation. Journal of Statistical Software, Jan. 2009
> [4] Robust mixture modelling using the t distribution. Statistics and Computing, 10(4):339–348, 2000.
> [5] A general and adaptive robust loss function. CVPR 2019
> [6] A decoder-only foundation model for time-series forecasting. ICML 2024
> [7] Chronos: Learning the language of time series. TMLR 2024
> [8] Time-MoE: Billion-Scale Time Series Foundation Models with Mixture of Experts. ICLR 2025
> [9] UniTS: A Unified Multi-Task Time Series Model. NeurIPS 2024.
> [10] PowerPM: Foundation Model for Power Systems. NeurIPS 2024.

---

> > ### Author Response · Authors · 2025-08-05
> >
> > Dear Reviewer WAXQ, we believe we have responded to and addressed the comments raised in your initial review. Please let us know if there are any remaining points you would like to further discuss.

---

> > > ### Comment · Reviewer_WAXQ · 2025-08-06
> > > **Official Comment by Reviewer WAXQ**
> > >
> > > Thank you for the detailed rebuttal, which has addressed most of my concerns. Considering the significance of a large-scale forecasting benchmark within the observability domain, I will raise my recommendation score to 4.

---

### Official Review · Reviewer_pfs4 · 2025-07-01

**Clarity:** 3
**Significance:** 3
**Originality:** 3
**Rating:** 5
**Confidence:** 4

**Summary:**

This paper presents TOTO, a decoder-only foundation model tailored for time series forecasting in the domain of observability metrics. The authors also introduce BOOM, a large-scale benchmark consisting of 350 million observations across 2,807 real-world multivariate time series collected from production monitoring systems. TOTO incorporates several architecture innovations—including causal patch-based normalization, proportional factorized attention, and a Student-T mixture model head—specifically designed to address the challenges posed by observability data such as high dimensionality, nonstationarity, and heavy-tailed distributions.

**Questions:**

see weaknesses

**Ethical Concerns:**

["NO or VERY MINOR ethics concerns only"]

**Final Justification:**

I appreciate the clarifications and additional empirical evidence provided regarding the generalization capability of the Toto model beyond the observability domain. The results on GIFT-Eval and LSF, as well as the efforts to address potential data distribution concerns, have addressed my main concerns. I have decided to increase my score.

**Limitations:**

yes

**Quality:**

3

**Strengths And Weaknesses:**

Strengths
- The paper is clearly written and logically structured, making it easy to follow the motivation, methodology, and experimental results.
- Substantial contributions: The work introduces two major contributions to the time series forecasting community
- Strong empirical results: TOTO achieves state-of-the-art performance on three benchmarks under zero-shot settings, and demonstrates competitive performance even in full-shot scenarios.
- Reproducibility and openness: The authors provide open access to the model weights, training and inference code, as well as the BOOM datasets.

Weaknesses
- Potential overgeneralization in scope: While the paper presents strong contributions in the domain of observability data, the title and framing suggest broader applicability to general time series forecasting.Since both the model design and benchmark (BOOM) are heavily tailored to observability workloads, it may be somewhat overstated to position this work as a general-purpose time series foundation model without more extensive validation beyond this domain.
- Although the authors state that there is no data leakage between the TOTO pretraining corpus and the BOOM benchmark due to environment isolation, both datasets are sourced from the same commercial observability platform. This raises concerns about distributional similarity. Since TOTO is pre-trained on large-scale observability data while competing foundation models are generally trained on more diverse or generic time series corpora, the performance advantage on BOOM may partially reflect domain alignment rather than purely architectural superiority.

---

> ### Author Rebuttal · Authors · 2025-07-30
>
> Thank you for your thoughtful comments!
>
> > Potential overgeneralization in scope: While the paper presents strong contributions in the domain of observability data, the title and framing suggest broader applicability to general time series forecasting. Since both the model design and benchmark (BOOM) are heavily tailored to observability workloads, it may be somewhat overstated to position this work as a general-purpose time series foundation model without more extensive validation beyond this domain.
>
> You correctly point out that the two main contributions of the paper, Toto and BOOM, are motivated by applications within the observability domain. And indeed, the architectural choices of Toto were designed with performance on observability data in mind.
>
> However, we *do* find strong empirical evidence that Toto generalizes well to other domains. To demonstrate this, in addition to BOOM, we also provide evaluation results in Section 5 on GIFT-Eval \[1\] and LSF \[2\]. These are both widely accepted, general-purpose, multi-domain benchmarks that are not primarily focused on observability. They include many time series examples from non-observability domains, including finance, weather, energy, natural science, sales, and transportation. **We feel that Toto’s state-of-the-art performance on these benchmarks clearly demonstrates Toto’s applicability outside of observability.**
>
> > Although the authors state that there is no data leakage between the TOTO pretraining corpus and the BOOM benchmark due to environment isolation, both datasets are sourced from the same commercial observability platform. This raises concerns about distributional similarity. Since TOTO is pre-trained on large-scale observability data while competing foundation models are generally trained on more diverse or generic time series corpora, the performance advantage on BOOM may partially reflect domain alignment rather than purely architectural superiority.
>
> While we did put significant effort into ensuring no data overlap between our training and test data by sourcing them from isolated environments, we acknowledge that there likely is some distributional similarity. That being said, as mentioned above, Toto also achieves state-of-the-art performance on GIFT-Eval and LSF, two general-purpose benchmarks containing time series primarily from domains outside of observability. As we show in Section 4.3 (“Statistical Characteristics”), these benchmarks are distributionally dissimilar from BOOM, suggesting that Toto’s strong performance is not solely due to in-domain evaluation.
>
> ### **Please do let us know if there are any remaining concerns that prevent you from increasing your score.**
>
> \[1\] Gift-eval: A benchmark for general time series forecasting model evaluation. arXiv preprint arXiv:2410.10393, 2024
> \[2\] Informer: Beyond efficient transformer for long sequence time-series forecasting. AAAI, 2024

---

> > ### Comment · Reviewer_pfs4 · 2025-08-04
> >
> > Thank you for your detailed and thoughtful rebuttal. The results on GIFT-Eval and LSF, as well as the efforts to address potential data distribution concerns, have addressed my main concerns. I have decided to increase my score.

---

### Official Review · Reviewer_cYnS · 2025-07-03

**Clarity:** 4
**Significance:** 4
**Originality:** 4
**Rating:** 5
**Confidence:** 1

**Summary:**

This manuscript introduces Toto, a 151‑million‑parameter, decoder‑only transformer tailored for multivariate time‑series forecasting, with a focus on observability data (e.g., telemetry). Toto is trained on a large, diverse corpus—4–10× larger than prior time‑series models—combining proprietary observability data, public datasets, and synthetic time‑series. To benchmark Toto, the authors construct BOOM, a comprehensive dataset composed of 2,807 real-world time series and totaling 350 million observations. They evaluate Toto against both BOOM and established benchmarks, demonstrating state‑of‑the‑art performance in forecasting accuracy.

**Questions:**

- Please explain the data split in BOOM, given the heterogeneity of the data how the split was handled? And does the split happen at data point level or query level?

- How do you ensure the quality of the labels generated from the LLM and verified by humans is consistent? Please elaborate on the labeling approach, and if any inter-annotator agreement (for multiple humans) has been deployed.

**Ethical Concerns:**

["NO or VERY MINOR ethics concerns only"]

**Final Justification:**

Authors responded to my questions, and my score is now final.

**Limitations:**

Yes.

**Paper Formatting Concerns:**

N/A.

**Quality:**

4

**Strengths And Weaknesses:**

The paper bridges the gap between foundation models and time‑series data by leveraging scale and architectural tuning.
Toto captures nuanced temporal patterns present in observability workloads by being trained on a massive mixed corpus - something that smaller, domain‑limited models often miss. The decoder‑only transformer design, alongside task‑specific innovations, shows a specific adaptation approach in the model design instead of a generic application of NLP architectures to time series. The construction of BOOM dataset is a major contribution. BOOM is a benchmark based on real-world data (not synthetic) on a scale rarely seen in forecasting research. By using over 2,800 traces and rigorous evaluation, the authors offer both transparency and a meaningful metric for future models. Authors also report comparisons with other models on three datasets (including BOOM) and share zero-shot and full-shot results. Toto consistently  beats other models and generalizes well to existing forecasting datasets, suggesting robustness and broader applicability.

---

> ### Author Rebuttal · Authors · 2025-07-30
>
> Thank you for the insightful comments. Below are the answers to your questions:
>
> > Please explain the data split in BOOM, given the heterogeneity of the data how the split was handled? And does the split happen at data point level or query level?
>
> We describe the evaluation setting for BOOM, including details of the train/test split, in Appendix C.2 in the supplemental materials. To summarize, we adopt the standard practice within the forecasting literature of splitting each time series along the time dimension. Following the same setting used by GIFT-Eval \[1\], we designate the first 90% of each series as train and the last 10% as test. We do this split per query, meaning that each variate within a multivariate time series has the train/test split at the same timestamp.
>
> Additional information regarding data preprocessing and quality controls for BOOM is provided in Appendix C.1.
>
> > How do you ensure the quality of the labels generated from the LLM and verified by humans is consistent? Please elaborate on the labeling approach, and if any inter-annotator agreement (for multiple humans) has been deployed.
>
> LLM labeling was used to identify the subdomains (“Application”, “Network”, etc.) of the time series in the BOOM benchmark. This labeling does not impact the main results in Table 2, but rather only affects the taxonomy-level statistics provided in Figure 4 and the breakdowns in Tables 8 and 14 in the appendices. To clarify, the labeling followed a two-step protocol:
>
> 1. LLM Pre-labeling: system domains were initially labeled by a large language model
> 2. Human Review: The labels were then manually reviewed by multiple experts with domain knowledge. Each expert received a disjoint subset of the dataset to review. The experts consulted each other when they felt a particular example was ambiguous.
>
> ### **Please do let us know if there are any remaining concerns that prevent you from increasing your score.**
>
> \[1\] Gift-eval: A benchmark for general time series forecasting model evaluation. arXiv preprint arXiv:2410.10393, 2024

---

### Note · Authors · 2025-08-11

Through our rebuttals we feel we have addressed most if not all outstanding concerns, and three of four reviewers have kindly indicated that they would raise their preliminary scores. We thank all our reviewers for their thoughtful engagement.

---

### Decision · Program_Chairs · 2025-09-17

**Decision:**

Accept (poster)

**Comment:**

This paper introduces TOTO, a foundation model for time series forecasting, addressing challenges in observational data . It also presents BOOM, a large-scale benchmark with 350 million observations. Experiments demonstrate strong performance on BOOM and GIFT-Eval.

The authors provided clear and thorough responses, highlighting TOTO’s advantages and fine-tuning improvements. Overall, this is a solid contribution, and I recommend acceptance.